# Blood flow guides sequential support of neutrophil arrest and diapedesis by PILR-β 1 and PILR-α

**Yu-Tung Li, Debashree Goswami[†], Melissa Follmer, Annette Artz, Mariana Pacheco-Blanco, Dietmar Vestweber\***

Vascular Cell Biology, Max Planck Institute of Molecular Biomedicine, Münster, Germany

**Abstract** Arrest of rapidly flowing neutrophils in venules relies on capturing through selectins and chemokine-induced integrin activation. Despite a long-established concept, we show here that gene inactivation of activating paired immunoglobulin-like receptor (PILR)-β1 nearly halved the efficiency of neutrophil arrest in venules of the mouse cremaster muscle. We found that this receptor binds to CD99, an interaction which relies on flow-induced shear forces and boosts chemokine-induced $\beta_2$-integrin-activation, leading to neutrophil attachment to endothelium. Upon arrest, binding of PILR-β1 to CD99 ceases, shifting the signaling balance towards inhibitory PILR-α. This enables integrin deactivation and supports cell migration. Thus, flow-driven shear forces guide sequential signaling of first activating PILR-β1 followed by inhibitory PILR-α to prompt neutrophil arrest and then transmigration. This doubles the efficiency of selectin-chemokine driven neutrophil arrest by PILR-β1 and then supports transition to migration by PILR-α.
DOI: https://doi.org/10.7554/eLife.47642.001

**\*For correspondence:**
vestweb@mpi-muenster.mpg.de

**Present address:** [†]Center for Global Infectious Disease Research, Seattle Childrens Research Institute, Seattle, United States

**Competing interests:** The authors declare that no competing interests exist.

## Introduction

Host defense against pathogens depends on the recruitment of leukocytes to sites of infections (*Ley et al., 2007*). Selectins capture leukocytes to the endothelial cell surface by binding to glyco-conjugates (*McEver, 2015*). This process is supported by shear forces caused by the blood stream, since selectin interactions with their ligands, different to most other protein-protein interactions, are improved, not impaired, by low tension forces. These interactions trigger, in combination with chemokines, signals that activate leukocyte integrins, such as LFA-1 and Mac-1 on neutrophils. Integrins then mediate leukocyte arrest, followed by crawling to appropriate exit sites and diapedesis (transmigration) through the vessel wall (*Nourshargh and Alon, 2014*; *Vestweber, 2015*).

CD99 is an O-glycosylated cell surface protein expressed on most leukocytes and endothelial cells, which participates in the diapedesis process. Antibodies against CD99 blocked the transmigration of monocytes in vitro (*Schenkel et al., 2002*) and lymphocyte and neutrophil extravasation in vivo (*Bixel et al., 2004*; *Bixel et al., 2007*; *Dufour et al., 2008*). More detailed analysis revealed that neutrophils accumulated between the endothelium and the basement membrane (*Bixel et al., 2007*; *Bixel et al., 2010*; *Watson et al., 2015*).

Recently, we confirmed these results by analyzing CD99 gene inactivated mice (*Goswami et al., 2017*). Unexpectedly, we also found that neutrophil arrest was strongly reduced in postcapillary venules of these mice. Furthermore, it turned out that leukocyte extravasation required CD99 only on endothelial cells but not on neutrophils, suggesting the relevance for a heterophilic ligand on neutrophils. In vitro, purified CD99 enhanced chemokine induced integrin activation on neutrophils. Potential candidates for relevant neutrophil receptors for this effect are the paired immunoglobulin-

like receptors α and β (PILR-α and PILR-β) which have been described as binding partners for CD99 (*Goswami et al., 2017*).

Like many other paired receptors of the immune system (*Yamada and McVicar, 2008*; *Barrow and Trowsdale, 2008*), PILR-α and PILR-β are structurally related membrane receptors that share similar extracellular regions but differ dramatically in the rest of their structure, which enables them to bind similar ligands, but to transmit either inhibitory or activating signals. The inhibitory receptor PILR-α bears two immunoreceptor tyrosine-based inhibitory motifs (ITIMs) in its cytoplasmic part. In contrast, activating PILR-β contains no cytoplasmic domain and associates through its transmembrane domain with the signaling adaptor DAP12 that contains an immunoreceptor tyrosine-based activation motif (ITAM).

Initially, CD99 on lymphocytes was discovered as ligand for PILR-α and PILR-β on dendritic cells (DC) and natural killer (NK) cells (*Shiratori et al., 2004*). PILRs bind to sialylated O-glycans on CD99 in conjunction with proteinaceous structural elements (*Wang et al., 2008*; *Sun et al., 2012*). Importantly, inhibitory PILR-α binds to CD99 with 40 times higher affinity than PILR-β (*Tabata et al., 2008*). The low binding affinity of PILR-β is one of the weakest among all known paired receptors, which raises the question whether it has a chance to bind CD99 if PILR-α is co-expressed on the same cell (*Kuroki et al., 2012*).

Neutrophils express both forms. Gene inactivation of the inhibitory PILR-α was reported to enhance neutrophil recruitment to inflamed peritoneum (*Wang et al., 2013*). However, involvement of endothelial CD99 is unclear. Furthermore, the extremely low affinity of PILR-β for CD99 also raised doubts about its relevance as receptor for CD99 during the extravasation process.

Here we have analyzed whether PILRs, and if so which ones, are indeed relevant in vivo for the contribution of CD99 to the process of leukocyte extravasation. To this end we have generated gene-inactivated mice for PILR-α and PILR-β. We present evidence that PILR-β is responsible for the CD99 induced enhancement of chemokine driven integrin activation. In addition, we found that the interaction of PILR-β on neutrophils with CD99 (on endothelial cells) is driven by flow-induced shear forces. We propose a mechanism by which flow enables endothelial CD99 to interact with PILR-β, which stimulates doubling of chemokine induced leukocyte arrest in inflamed venules. Subsequently, PILR-α is needed as negative modulator of integrin activity to support the transition to leukocyte diapedesis.

## Results

### Generation and characterization of mice gene-inactivated for PILR-α and PILR-β1

To evaluate a role of PILR-α or PILR-β in leukocyte extravasation, gene inactivated mice were generated by the CRISPR-Cas9 approach (Material and methods). Analyzing PILR-α$^{-/-}$ mice, absence of PILR-α protein expression was confirmed by flow cytometry (FACS) of permeabilized neutrophils using PILR-α-specific antibodies against the C-terminus (*Figure 1A*). Surface expression of PILR-β, LFA-1, CD11b, CD11c, PSGL-1 and CXCR-2 was unaltered (*Figure 1A and B*).

To compare the expression levels of PILR proteins on neutrophils, we stained WT and PILR-α$^{-/-}$ PMN with antibodies against PILR-β-Fc, which were cross-reactive for the highly similar extracellular domains of both PILR subtypes. Staining of PILR-α$^{-/-}$ neutrophils was strongly reduced compared to WT neutrophils and specific signals were completely blocked with the PILR-β-Fc fusion protein (*Figure 1—figure supplement 1A and B*). Quantifying mean fluorescence intensity (MFI) of two similar experiments revealed 62% (±2%) of the total FACS signal being due to PILR-α (*Figure 1—figure supplement 1C*).

PILR-β1$^{-/-}$ neutrophils showed an 80% reduced MFI signal with PILR-β-specific antibodies compared to WT neutrophils (*Figure 1C and D*). Expression levels of PILR-α, LFA-1, CD11b, CD11c, PSGL-1 and CXCR-2 were normal. The PILR-β locus contains two closely adjacent genes for PILR-β1 and PILR-β2 with 92% sequence homology. Quantitative RT-PCR revealed that our PILR-β targeted mice were specifically gene inactivated for PILR-β1 and still expressed PILR-β2 (*Figure 1E*). According to our FACS analysis PILR-β2 represents 20% of the overall expression level of the two PILR-β genes.

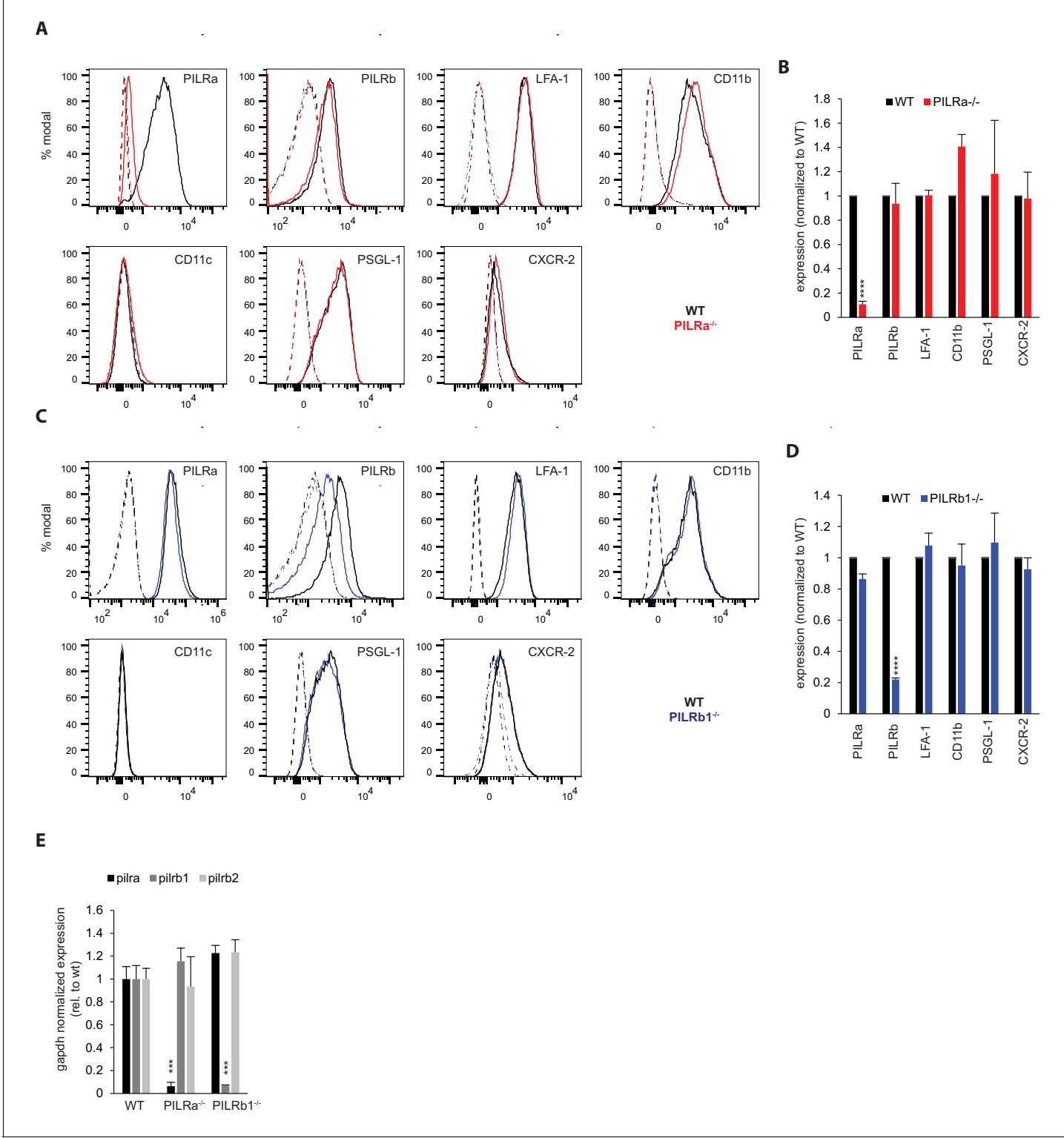

**Figure 1.** Characterization of PILR-α$^{-/-}$ and PILR-β1$^{-/-}$ neutrophils for surface expression of molecules involved in extravasation. Ly6G$^{+}$ bone marrow neutrophils from WT, PILR-α$^{-/-}$ or PILRβ1$^{-/-}$ mice were analyzed for indicated antigens by FACS. Representative histograms of at least three independent experiments are shown in (**A**) for PILR-α$^{-/-}$ and (**C**) for PILR-β1$^{-/-}$ mice, with expression levels quantified in (**B**) and (**D**), respectively. Solid tracks, specific staining; dotted tracks, isotype control. Groups were compared by 2-tailed t-test. (**E**) qRT-PCR analysis of RNA extracted from peripheral blood of WT (n = 9) or PILR-α$^{-/-}$ (n = 5) or PILR-β1$^{-/-}$ (n = 7) mice for the indicated genes. Groups were analyzed by 1-way ANOVA followed by Holm-Sidak method for multiple comparisons. Error bars, SEM. ****p<0.001.

*Figure 1 continued on next page*

*Figure 1 continued*

DOI: https://doi.org/10.7554/eLife.47642.002

The following source data and figure supplements are available for figure 1:

**Source data 1.** Source data for *Figure 1B and D*.
DOI: https://doi.org/10.7554/eLife.47642.010

**Source data 2.** Source data for *Figure 1E*.
DOI: https://doi.org/10.7554/eLife.47642.011

**Figure supplement 1.** Quantification of PILR subtypes neutrophils.
DOI: https://doi.org/10.7554/eLife.47642.003

**Figure supplement 1—source data 1.** Source data for *Figure 1—figure supplement 1B-C*.
DOI: https://doi.org/10.7554/eLife.47642.004

**Figure supplement 2.** Characterization of immune subsets in PILR-$\alpha^{-/-}$ and PILR-$\beta1^{-/-}$mice.
DOI: https://doi.org/10.7554/eLife.47642.005

**Figure supplement 2—source data 1.** Source data for *Figure 1—figure supplement 2A*.
DOI: https://doi.org/10.7554/eLife.47642.006

**Figure supplement 2—source data 2.** Source data for *Figure 1—figure supplement 2B* (F4/80+).
DOI: https://doi.org/10.7554/eLife.47642.007

**Figure supplement 2—source data 3.** Source data for *Figure 1—figure supplement 2B* (Ly6G+).
DOI: https://doi.org/10.7554/eLife.47642.008

**Figure supplement 2—source data 4.** Source data for *Figure 1—figure supplement 2C*.
DOI: https://doi.org/10.7554/eLife.47642.009

Both gene inactivated mouse lines showed normal peripheral leukocyte composition (*Figure 1—figure supplement 2*). Off-target effects were excluded by sequencing of potential genes (Material and methods).

## PILR-α and PILR-β1 exert different functions in neutrophil extravasation

In order to test whether PILR-α and PILR-β1 participate in leukocyte extravasation and whether rolling, adhesion or transmigration is affected, mice were challenged intrascrotally by TNF-α for 2 hr followed by intravital microscopy (IVM) of the cremaster muscle. Neutrophil rolling flux, defined as the percentage of all leukocytes traveling through the microvessels, remained unaffected in either PILR knockout mouse line (*Figure 2A*). However, rolling velocity was enhanced in PILR-$\beta1^{-/-}$ mice compared to PILR-$\alpha^{-/-}$ mice (*Figure 2B*). Neutrophils lacking PILR-α exhibited a small, but not significant increase in adhesion and extravasation. In contrast, PILR-β1 deficiency caused significant reduction of both adhesion and extravasation by 40% (±6%) and 31% (±5%), respectively (*Figure 2C and D*). Representative videos for the IVM analysis of mice of all three genotypes are given in the *Videos 1–3*. Thus, PILR-β1 is needed for optimal levels of adhesion, rolling velocity and extravasation of neutrophils in vivo.

## PILR-β1 is required for CD99 support of chemokine-induced neutrophil binding to ICAM-1

We have previously reported that purified immobilized CD99 enhances the binding of neutrophils to immobilized ICAM-1 under flow, when CXCL1 and P-selectin were co-coated (*Goswami et al., 2017*). To clarify a potential role of the PILRs in these effects, we used primary isolated PILR-$\alpha^{-/-}$ and PILR-$\beta1^{-/-}$ neutrophils in similar experiments. Competitive adhesion assays under flow conditions were performed with 1:1 mixtures of fluorescence-labeled WT and unlabeled mutant neutrophils. The flow chamber surface was coated with P-selectin-Fc, ICAM-1-Fc, and CXCL-1, and either CD99-Fc or a control IgG. The number of arrested neutrophils did not essentially differ between WT and either knockout of PILR on the control IgG co-coated surface. CD99-Fc significantly enhanced arrest of WT and PILR-$\alpha^{-/-}$ neutrophils (*Figure 3A*), but not of PILR-$\beta1^{-/-}$ neutrophils (*Figure 3B*), indicating that PILR-β1 is required for CD99-enhanced chemokine induced neutrophil arrest under flow on ICAM-1.

To test directly, whether ligation of PILR would enhance chemokine-induced activation of β2-integrins, we pre-stimulated PILRs on WT, PILR-$\alpha^{-/-}$ and PILR-$\beta1^{-/-}$ neutrophils with antibodies and

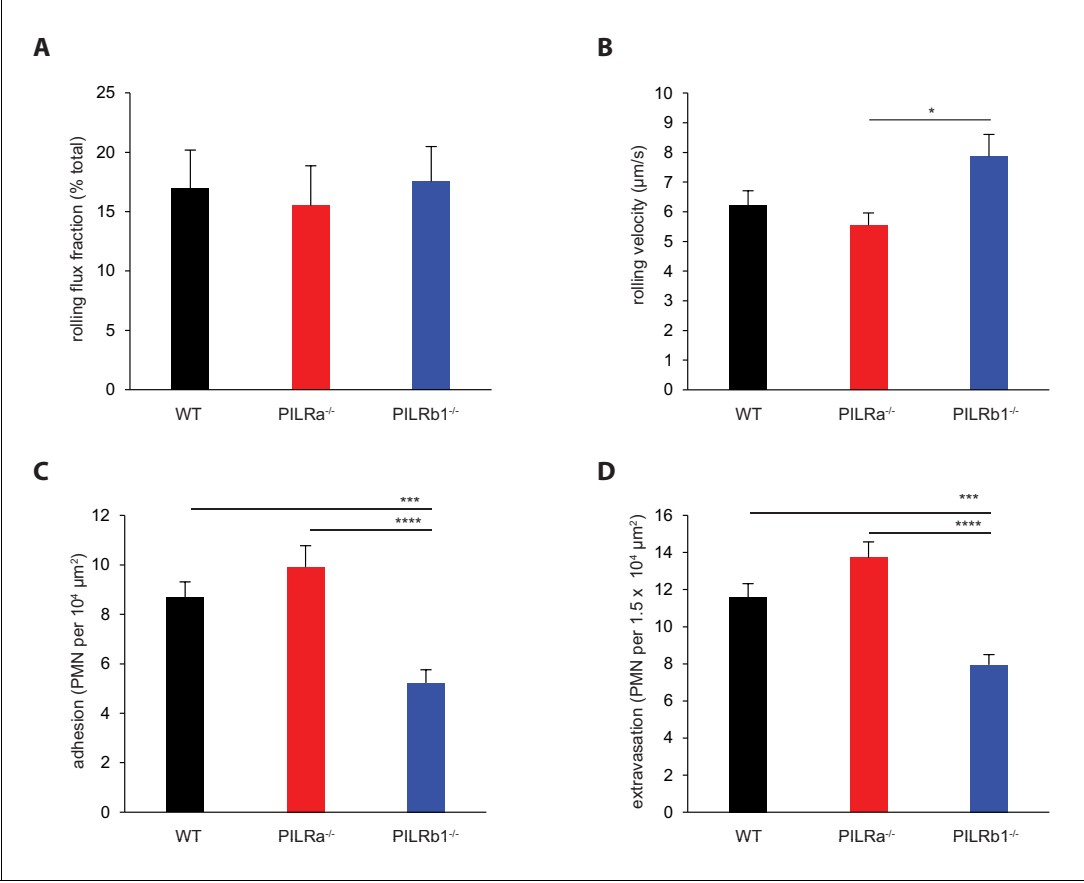

**Figure 2.** Impact of PILRs on neutrophil extravasation in the inflamed cremaster. WT, PILR-α⁻/⁻ and PILR-β1⁻/⁻ mice were stimulated with intrascrotal TNF-α for 2 hr. Cremaster muscles were exposed and examined for neutrophil extravasation by intravital microscopy (IVM) for (**A**) rolling flux fraction, (**B**) rolling velocity, (**C**) adherent leukocytes, and (**D**) extravasated leukocytes. WT: n = 40 vessels from eight mice; PILR-α⁻/⁻: n = 37 vessels from seven mice; PILR-β1⁻/⁻: n = 35 vessels from seven mice. For (**B**), 117 cells (WT), 111 cells (PILR-α⁻/⁻) and 105 cells (PILR-β1⁻/⁻) from the indicated number of vessels were quantified. Groups were analyzed by 1-way ANOVA followed by Tukey's multiple comparisons. Error bars, SEM. *p<0.05, ***p<.005, ****p<0.001. Hemodynamic parameters are given in **Table 1**.

DOI: https://doi.org/10.7554/eLife.47642.012
The following source data is available for figure 2:

**Source data 1.** Source data for **Figure 2A, C and D**.
DOI: https://doi.org/10.7554/eLife.47642.013
**Source data 2.** Source data for **Figure 2B**.
DOI: https://doi.org/10.7554/eLife.47642.014
**Source data 3.** Source data for **Figure 2A–D** (**Table 1**).
DOI: https://doi.org/10.7554/eLife.47642.015

determined chemokine-induced β2-integrin activation by binding of soluble ICAM-1-Fc. To this end, we incubated bone marrow cells of the different genotypes (in the presence of ICAM-1-Fc) with polyclonal anti PILR antibodies (recognizing both PILR isoforms) for 15 min, followed by 0, 3 and 5 min incubation with CXCL1. After fixation, Ly6G⁺ cells were analyzed for ICAM-1-Fc binding by FACS. For all genotypes, PILR crosslinking before chemokine stimulation enhanced ICAM-1-binding only weakly. For WT neutrophils, PILR crosslinking strongly enhanced ICAM-1 binding by 48% (±6%) at 3 and 5 min post-chemokine stimulation (**Figure 3C**). Slightly stronger increases were found for PILR-α⁻/⁻ neutrophils (76% (±8%)) (**Figure 3D**). In contrast, the increase of integrin activation was strongly reduced to only 18% (±3%) for PILR-β1⁻/⁻ neutrophils (**Figure 3E**). The residual increase could be due to the presence of PILR-β2. When a calcium ionophore instead of CXCL-1 was used to stimulate WT neutrophils, PILRs crosslinking no longer enhanced ICAM-1 binding (**Figure 3F**). Since calcium influx is downstream of chemokine induced GPCR signaling (**Block et al., 2016**), we

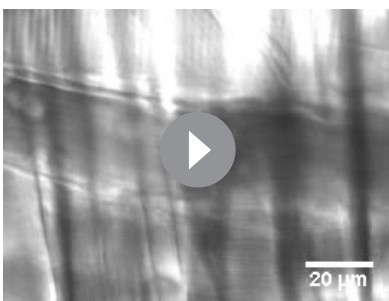

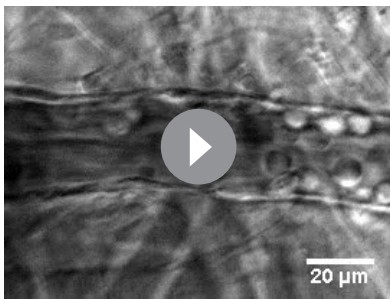

**Video 1.** Representative IVM video of a vessel in WT in *Figure 2*.
DOI: https://doi.org/10.7554/eLife.47642.017

**Video 2.** Representative IVM video of a vessel in PILR-$\alpha^{-/-}$ in *Figure 2*.
DOI: https://doi.org/10.7554/eLife.47642.018

conclude that PILR-β1 acts at a step upstream of calcium influx. Collectively, these in vitro findings suggest PILR-β1 on neutrophils is required for CD99 stimulated signals that enhance chemokine-induced β2 integrin activation.

To learn more about the signaling mechanism behind these effects we tested whether the tyrosine kinase Syk could be involved. We found that chemokine induced tyrosine phosphorylation of Syk was strongly enhanced if PMNs were co-incubated with polyclonal antibodies recognizing the extracellular domain of both PILR forms (*Figure 3—figure supplement 1A*). To test the relevance of Syk for the modulating effect of PILR-α and PILR-β1 on the adhesive function of β$_2$-integrins, we performed adhesion assays with immobilized ICAM-1-Fc and chemokine-stimulated PMNs of all three genotypes treated with or without the Syk inhibitor PRT-060318. We found that the Syk inhibitor blocked the adhesion modulating effects of each of the two PILR receptors, (i.e. adhesion of the KO cells was as efficient as that of WT cells if all cells were treated with the inhibitor) (*Figure 3—figure supplement 1B*). To investigate the involvement of Syk in CD99 facilitated chemokine-induced adhesion, PILR-α$^{-/-}$ or PILR-β1$^{-/-}$ neutrophils were allowed to adhere under flow competitively with WT neutrophils on a surface coated with P-selectin-Fc, CXCL-1, ICAM-1-Fc and CD99-Fc or a control IgG as described in the presence of the Syk inhibitor. We observed that the adhesion support by CD99 for WT and PILR-α$^{-/-}$ neutrophils vanished when Syk was inhibited (*Figure 3G and H*), clearly indicating that CD99 driven β2 integrin activation enhancement requires Syk activity.

## PILR-β1, but not PILR-α, is required for CD99-support of neutrophil arrest in vivo

To test, whether CD99 mediated support of neutrophil arrest in vivo depends on PILR-β1, we generated chimeric mice by bone marrow transplantation, which were deficient for CD99 on endothelium and lacked one of the two PILRs on neutrophils. Neutrophil extravasation was stimulated and analyzed by IVM as for *Figure 2*. In agreement with our previous findings (*Goswami et al., 2017*), ablation of endothelial CD99 reduced adhesion of WT neutrophils by 57% (*Figure 4B*) and 43% (*Figure 4E*). Neutrophils lacking PILR-β1 and transplanted into WT recipients phenocopied this effect (46 ± 8%) (*Figure 4E*). Importantly, the combination of PILR-β1$^{-/-}$ neutrophils and CD99$^{-/-}$ endothelium did not further reduce neutrophil arrest (34 ± 8%) (*Figure 4E*), suggesting both proteins act in the same pathway. In contrast, neutrophils lacking PILR-α were more adhesive than their WT counterparts (38 ± 8%) (*Figure 4B*). In general, all adhesion effects observed in the various chimeric mice translated into extravasation effects, with the exception of PILR-α gene deficiency (*Figure 4B and C*), where extravasation was like in WT mice despite increased numbers of arrested neutrophils. This may point to a PILR-α function downstream of adhesion (see

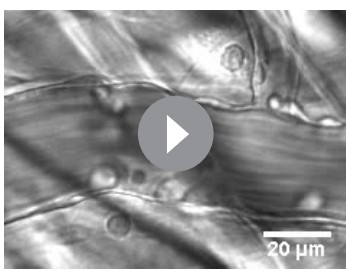

**Video 3.** Representative IVM video of a vessel in PILR-β1$^{-/-}$ in *Figure 2*.
DOI: https://doi.org/10.7554/eLife.47642.019

**Table 1.** Hemodynamic parameters of mice in full knockout characterization in **Figure 2**.
Genotypes, number of mice, number of venules, venule diameters, leukocyte counts, blood velocities and Newtonian wall shear stresses are shown as mean ± SEM.

| Genotype | Mice | Venules | Diameter (μm) | Leukocyte counts ($10^6$ cells/ml) | Mean blood velocity (mm/s) | Newtonian wall shear rate ($s^{-1}$) |
|---|---|---|---|---|---|---|
| WT | 8 | 40 | 28.7 ± 0.7 | 4.34 ± 0.62 | 1.22 ± 0.02 | 350 ± 9 |
| PILRα$^{-/-}$ | 7 | 37 | 29.4 ± 0.8 | 5.19 ± 0.46 | 1.23 ± 0.02 | 343 ± 9 |
| PILRβ1$^{-/-}$ | 7 | 35 | 28.2 ± 0.8 | 3.95 ± 0.45 | 1.23 ± 0.02 | 357 ± 10 |

DOI: https://doi.org/10.7554/eLife.47642.016

below). In none of the various chimeric mice, rolling flux was significantly affected (*Figure 4A and D*).

## CD99 interaction with PILR-β1 is induced by shear forces

The binding affinity between CD99 and PILR-β is rather low (2.2 μM) and 40-fold weaker than for the PILR-α counterpart (*Tabata et al., 2008*). This raises the question, how endothelial CD99 manages to interact with PILR-β. Since neutrophil adhesion in vessels occurs under flow, we tested whether flow-induced shear forces modulate the CD99-PILR interactions. Therefore, we studied the interaction of CD99-tranfected CHO cells (*Figure 5A*), with PILR-α-Fc or PILR-β-Fc coated surfaces in the absence or presence of flow. Under static conditions, 22 fold more CHO-CD99 cells adhered to PILR-α-Fc than to PILR-β-Fc (*Figure 5B*).

Using an alternative flow assay, we video-recorded transiently interacting (≥50 ms) CHO-CD99 passing through either uncoated or PILR-α-Fc or PILR-β-Fc coated flow chambers under conditions of a continuously increasing shear gradient from 0.1 to 0.7 dyn/cm$^2$ (*Figure 5E*). A peak of specific interactions was observed between 0.1 and 0.3 dyn/cm$^2$ for both PILR-α-Fc- and PILR-β-Fc-coated surfaces, with the latter clearly being more supportive (*Figure 5F*). These PILR-α/β-Fc specific interactions were specific for CD99 under a constant shear of 0.2 dyn/cm$^2$ (*Figure 5—figure supplement 1*).

To extend our interaction studies under flow to neutrophils with their endogenous PILR expression levels, we passed primary isolated neutrophils of different genotype through flow chambers coated either with CD99-Fc or with control IgG, under the same experimental conditions as we applied before (*Figure 5E*). Transiently interacting neutrophils (≥30 ms) were videotaped. WT and also PILR-α$^{-/-}$ neutrophils interacted most efficiently with the CD99-Fc coated surfaces at ~0.2 dyn/cm$^2$, whereas this shear driven interaction was lost with PILR-β1$^{-/-}$ neutrophils (*Figure 5G*). Thus, neutrophil interactions with CD99 under flow are exclusively mediated by PILR-β1. Collectively, our results suggest that shear stress improves the binding of CD99 to PILR-β1, which compensates the inefficient binding characteristics at static conditions.

## Shear stress is needed for PILR-β1 to support neutrophil arrest

To extend our in vitro studies beyond the use of purified, immobilized proteins, we analyzed the interaction of WT and PILR-β1$^{-/-}$ neutrophils with primary isolated mouse dermal microvascular endothelial cells (MDMVEC) under static and flow conditions. MDMVEC were stimulated with TNF-α and CXCL1. Under static conditions, adhesion of PILR-β1$^{-/-}$ neutrophils was insignificantly different from WT neutrophils (*Figure 6B*). Comparable results were found if the shear gradient was reversed (*Figure 6C*) or shear was kept constant (*Figure 6D*). In contrast, under flow conditions with continuously increasing shear from 0 to 2 dyn/cm$^2$, PILR-β1$^{-/-}$ cells adhered 2.8-fold less efficient than WT cells (*Figure 6B and C*). Thus, a significant contribution of PILR-β1 to neutrophil attachment on inflamed endothelial cells is only detectable under flow conditions.

## PILR-α supports crawling and diapedesis

Analyzing the extravasation of PILR-α$^{-/-}$ neutrophils in vivo by intravital microscopy revealed that attachment to the vessel wall was enhanced, whereas extravasation was unaffected (*Figure 4*). Increased cell attachment was also prominent in in vitro adhesion assays with PILR-α$^{-/-}$ neutrophils and TNF-α stimulated MDMVEC (*Figure 7—figure supplement 1*). To clarify, why enhanced

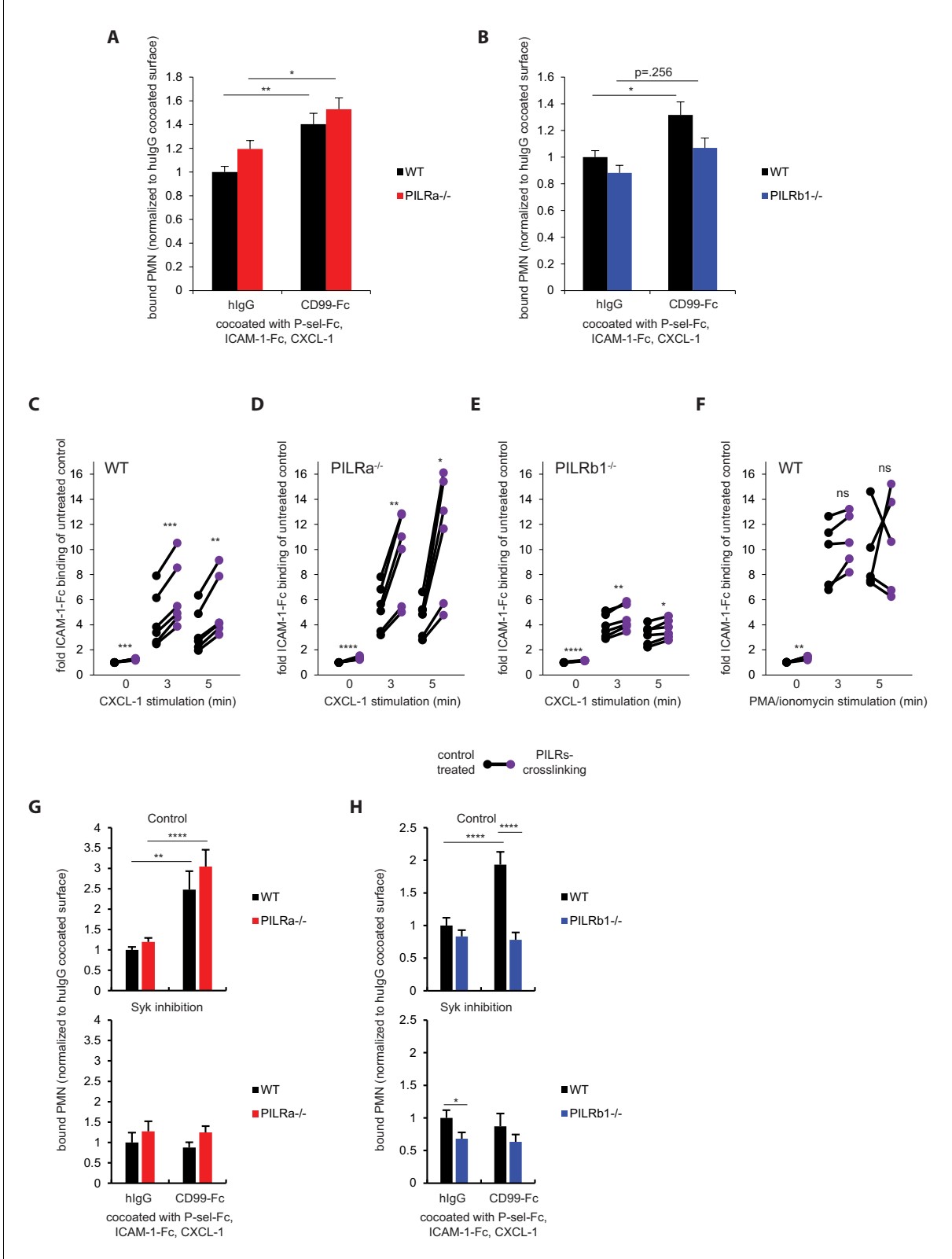

**Figure 3.** PILR-β1 is required for CD99-stimulated neutrophil adhesion on ICAM-1-Fc in vitro. Flow chamber surfaces were coated with P-selectin-Fc, ICAM-1-Fc and CXCL-1, and co-coated with either CD99-Fc or human IgG1 control (hIgG). PILR-$\alpha^{-/-}$ (A) or PILR-$\beta1^{-/-}$ (B) PMN were mixed with fluorescently labeled WT PMN in 1:1 ratio and passed over the surfaces at 5 dyn/cm$^2$ at 37°C for 5 min. Adherent PMNs were quantified for n = 45 fields from nine experiments each. (C–E) Bone marrow cells from WT (C), PILR-$\alpha^{-/-}$ (D) or PILR-$\beta1^{-/-}$ (E) mice were mixed with ICAM-1-Fc/PE-conjugated

*Figure 3 continued on next page*

*Figure 3 continued*

secondary antibody detection complex. Cell mixtures were pre-treated with non-crosslinking rabbit IgG or crosslinking polyclonal anti-PILRs in pair for 15 min, and then stimulated with 100 ng/ml CXCL-1 for 0 min (unstimulated), 3 and 5 min, followed by fixation and analysis for ICAM-1-Fc binding to Ly6G$^+$ cells by FACS. n = 6 paired samples per treatment per genotype. (F) Experiment described in (C) was repeated except for replacing the chemokine stimulation by 20 ng/ml PMA + 1 μg/ml ionomycin. n = 5 paired samples per treatment. (G–H) Experiments described in (A–B) were repeated in the presence of 1 μM Syk inhibitor PRT-060318 or vehicle control with PILR-α$^{-/-}$ (G) or PILR-β1$^{-/-}$ (H) neutrophils. n = 26 fields each from two experiments (G, control) or 39 fields each from three experiments (G, inhibition) fields. n = 20 fields each from two experiments (H, control) or 30 fields each from three experiments (H, inhibition) fields. Groups were compared by 1-way ANOVA followed by Tukey's multiple comparisons in (A, B, G, H) or 2-tailed paired t-test in (C–F). Error bars, SEM. *p<0.05, **p<.005, ***p<0.0005, ****p<0.0001.
DOI: https://doi.org/10.7554/eLife.47642.020

The following source data and figure supplements are available for figure 3:

**Source data 1.** Source data for *Figure 3A*.
DOI: https://doi.org/10.7554/eLife.47642.025
**Source data 2.** Source data for *Figure 3B*.
DOI: https://doi.org/10.7554/eLife.47642.026
**Source data 3.** Source data for *Figure 3C-E*.
DOI: https://doi.org/10.7554/eLife.47642.027
**Source data 4.** Source data for *Figure 3F*.
DOI: https://doi.org/10.7554/eLife.47642.028
**Source data 5.** Source data for *Figure 3G*.
DOI: https://doi.org/10.7554/eLife.47642.029
**Source data 6.** Source data for *Figure 3H*.
DOI: https://doi.org/10.7554/eLife.47642.030
**Figure supplement 1.** PILRs modulate β2 integrin activity via Syk.
DOI: https://doi.org/10.7554/eLife.47642.021
**Figure supplement 1—source data 1.** Source data for *Figure 3—figure supplement 1A*.
DOI: https://doi.org/10.7554/eLife.47642.022
**Figure supplement 1—source data 2.** Source data for *Figure 3—figure supplement 1B* (PILRa-/-).
DOI: https://doi.org/10.7554/eLife.47642.023
**Figure supplement 1—source data 3.** Source data for Source data for *Figure 3—figure supplement 1B* (PILRb1-/-).
DOI: https://doi.org/10.7554/eLife.47642.024

adhesion does not lead to more transmigration, we tested whether PILR-α would participate in the transmigration steps downstream of the adhesion step. Therefore, we performed static transmigration assays through primary MDMVEC with PILR-α$^{-/-}$, PILR-β1$^{-/-}$, and WT neutrophils. In order to compare transmigration without the complication of enhanced adhesive function of PILR-α$^{-/-}$ neutrophils, assays were performed with neutrophil numbers ranging from 1 to 15 × 10$^5$ cells per transwell filter. After 40 min, transmigrated neutrophils were counted, in pair with counting neutrophils adherent to the apical surface of MDMVEC. Results were divided into four groups according to adhesion levels and numbers of transmigrated cells were plotted for each group and each genotype (*Figure 7A*). Whereas WT neutrophils and PILR-β1$^{-/-}$ neutrophils transmigrated with similar efficiency when standardized to the same number of adherent cells, PILR-α$^{-/-}$ neutrophils clearly transmigrated less efficiently. This argued for a supportive role of PILR-α in the transmigration process at a step subsequent to the adhesion step. To verify this, we performed transmigration assays with WT and PILR-α$^{-/-}$ neutrophils for different time spans ranging from 15 to 90 min. At 45 and 60 min fewer PILR-α$^{-/-}$ than WT neutrophils had transmigrated (*Figure 7B*). At later time points, the number of transmigrating WT neutrophils plateaued and the PILR-α$^{-/-}$ neutrophils caught up. Thus, lack of PILR-α slows down transmigration.

To further clarify which step downstream of cell attachment would be supported by PILR-α during transmigration, we video-recorded crawling and diapedesis of WT and PILR-α$^{-/-}$ neutrophils passing under flow over TNF-α stimulated WT MDMVEC. The percentage of adherent cells that started to crawl (crawling initiation rate) and the time span required for each neutrophil to transmigrate through MDMVEC were determined. We found a 45% reduced crawling initiation rate of PILR-α$^{-/-}$ neutrophils compared to WT neutrophils (*Figure 7C*). In addition, the time needed for diapedesis was 2-fold increased in PILR-α$^{-/-}$ neutrophils (*Figure 7D*). Thus, PILR-α participates in transmigration

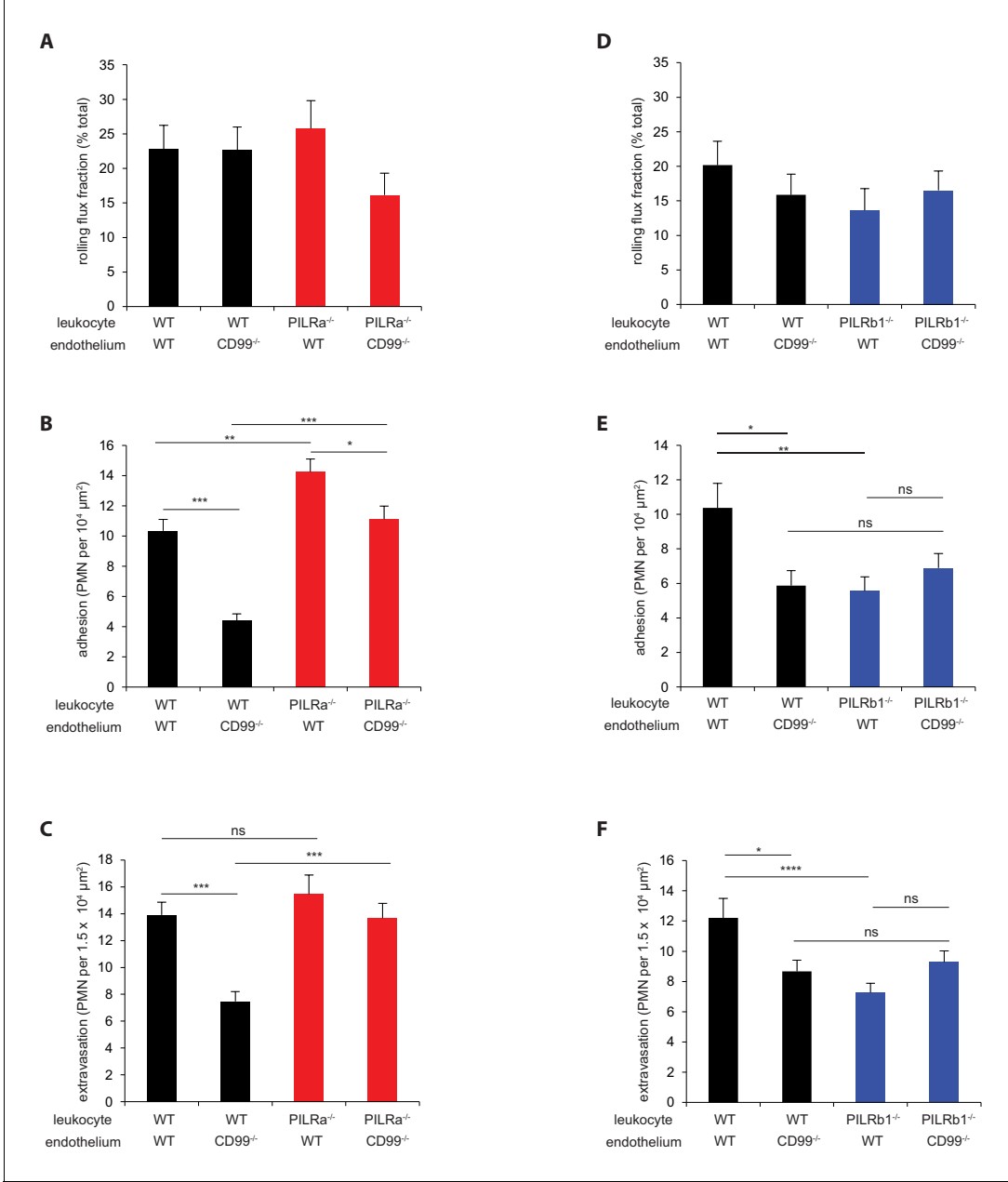

**Figure 4.** PILR-β1, but not PILR-α, is essential for CD99-stimulated support of neutrophil adhesion in vivo. WT or CD99$^{-/-}$ mice received bone marrow transplantation from WT (**A–F**) or PILR-α$^{-/-}$ (**A–C**) or PILR-β1$^{-/-}$ (**D–F**). IVM measurement of (**A, D**) rolling flux fraction, (**B, E**) adhesion and (**C, F**) extravasation were performed on TNF-α inflamed cremasters of the chimeric mice. Donor-Recipient: For (**A–C**), WT-WT, n = 39 vessels from five mice; WT-CD99$^{-/-}$, n = 38 vessels from five mice; PILR-α$^{-/-}$-WT, n = 39 vessels from five mice; PILR-α$^{-/-}$-CD99$^{-/-}$, n = 34 vessels from four mice. For (**D–F**), WT-WT, n = 23 vessels from four mice; WT-CD99$^{-/-}$, n = 28 vessels from five mice; PILR-β1$^{-/-}$-WT, n = 29 vessels from five mice; PILR-β1$^{-/-}$-CD99$^{-/-}$, n = 36 vessels from five mice. Groups were analyzed by 1-way ANOVA followed by Tukey's multiple comparisons. Error bars, SEM. *p<0.05, **p<0.01, ***p<.005. Hemodynamic parameters are given in *Tables 2* and *3*.
DOI: https://doi.org/10.7554/eLife.47642.031

The following source data is available for figure 4:

**Source data 1.** Source data for *Figure 4A–C*.
DOI: https://doi.org/10.7554/eLife.47642.032
**Source data 2.** Source data for *Figure 4A–C* (*Table 2*).
DOI: https://doi.org/10.7554/eLife.47642.033
**Source data 3.** Source data for *Figure 4D–F*.
DOI: https://doi.org/10.7554/eLife.47642.034
*Figure 4 continued on next page*

*Figure 4 continued*

**Source data 4.** Source data for *Figure 4D–F* (*Table 3*).

DOI: https://doi.org/10.7554/eLife.47642.035

by supporting the switch from leukocyte arrest to crawling and by supporting the efficiency of the diapedesis process.

## Discussion

Here, we show that the two antagonistic, paired receptors PILR-α and PILR-β1 each support neutrophil extravasation at different, consecutive steps of the extravasation process. Gene deficiency of the activating PILR-β1 receptor strongly impaired neutrophil arrest, which matched a similar effect we found recently in mice lacking endothelial CD99 (*Goswami et al., 2017*). In agreement with this, we show that despite a 40 times lower affinity of PILR-β1 for CD99 compared to PILR-α under static conditions (*Tabata et al., 2008*), flow-induced shear forces enhanced the interaction between CD99 and PILR-β1 and made it even superior over the interaction with PILR-α. Furthermore, in vitro flow adhesion assays revealed that PILR-β1 was required for CD99 to boost chemokine-induced activation of β2-integrins on neutrophils. Thus, PILR-β1 acts as a neutrophil receptor required for endothelial CD99 to support efficient neutrophil arrest. Gene inactivation of PILR-α affects the next step leading to reduced numbers of crawling neutrophils and impaired transmigration. We conclude that neutrophils require the inhibitory effect of PILR-α to redeem cellular motility by restraining β2-integrin activity. Collectively, we describe a molecular mechanism whereby flow-driven shear forces guide sequential signaling of first activating PILR-β1 followed by inhibitory PILR-α to prompt neutrophil arrest and subsequently transmigration (*Figure 8*).

Blood flow-induced shear forces largely prevent the interaction of leukocytes with the vessel wall keeping extravasation in healthy tissues at bay. The selectin/chemokine/integrin-based machinery explains, how inflamed venules circumvent this block of extravasation and even take advantage of shear forces as a driver for extravasation. Selectins on inflamed endothelial cells are specialized to capture leukocytes under flow, since their ligand-binding is improved under flow conditions (*Marshall et al., 2003*; *Yago et al., 2004*). Moreover, selectins trigger signals that activate integrins in addition to the chemokine stimulated signaling cascades, which finally leads to leukocyte arrest and attachment (*Zarbock et al., 2007*; *Block et al., 2012*). This model has been well studied over decades and is solidly established (*McEver and Zhu, 2010*; *Zarbock et al., 2011*). We suggest here, that CD99 and PILR-β1 represent new players in addition to selectins, chemokines and integrins in the multistep cascade of leukocyte arrest. Although selectins and chemokines trigger leukocyte integrin activation and leukocyte attachment efficiently in vivo, lack of CD99 or PILR-β1 reduces this efficiency by about twofold. This is a significant contribution. It is intriguing that similar to the selectins, the PILR-β1/CD99 interaction is also lectin-carbohydrate based and is supported by shear forces.

In light of the important role of PILR-β1 for leukocyte arrest and attachment, it is particularly interesting to consider how this is coordinated with the function of the inhibitory PILR-α. As mentioned

**Table 2.** Hemodynamic parameters of bone marrow transplanted chimeric mice for intravital microscopy in *Figure 4A–C*. Genotypes of donors and recipients, number of mice, number of venules, venule diameters, leukocyte counts, blood velocities and Newtonian wall shear stresses are shown as mean ± SEM.

| Donor genotype (neutrophil) | Recipient genotype (endothelium) | Mice | Venules | Diameter (μm) | Leukocyte counts ($10^6$ cells/ml) | Mean blood velocity (mm/s) | Newtonian wall shear rate ($s^{-1}$) |
|---|---|---|---|---|---|---|---|
| WT | WT | 5 | 39 | 25.4 ± 0.6 | 3.27 ± 0.19 | 1.21 ± 0.00 | 388 ± 8 |
| WT | CD99$^{-/-}$ | 5 | 38 | 24.2 ± 0.5 | 3.52 ± 0.07 | 1.20 ± 0.01 | 404 ± 8 |
| PILRα$^{-/-}$ | WT | 5 | 39 | 24.5 ± 0.6 | 2.87 ± 0.11 | 1.20 ± 0.01 | 403 ± 9 |
| PILRα$^{-/-}$ | CD99$^{-/-}$ | 4 | 34 | 23.8 ± 0.6 | 3.47 ± 0.12 | 1.21 ± 0.01 | 413 ± 10 |

DOI: https://doi.org/10.7554/eLife.47642.036

**Table 3.** Hemodynamic parameters of bone marrow transplanted chimera mice for intravital microscopy in *Figure 4D–F*. Genotypes of donors and recipients, number of mice, number of venules, venule diameters, leukocyte counts, blood velocities and Newtonian wall shear stresses are shown as mean ± SEM.

| Donor genotype (neutrophil) | Recipient genotype (endothelium) | Mice | Venules | Diameter (µm) | Leukocyte counts ($10^6$ cells/ml) | Mean blood velocity (mm/s) | Newtonian wall shear rate ($s^{-1}$) |
|---|---|---|---|---|---|---|---|
| WT | WT | 4 | 23 | 23.3 ± 0.7 | 3.13 ± 0.30 | 1.24 ± 0.01 | 431 ± 10 |
| WT | CD99$^{-/-}$ | 5 | 27 | 23.4 ± 0.8 | 5.46 ± 0.38 | 1.24 ± 0.01 | 434 ± 12 |
| PILRβ1$^{-/-}$ | WT | 5 | 29 | 24.0 ± 0.7 | 4.56 ± 0.88 | 1.23 ± 0.01 | 416 ± 10 |
| PILRβ1$^{-/-}$ | CD99$^{-/-}$ | 5 | 36 | 22.6 ± 0.5 | 5.58 ± 0.37 | 1.25 ± 0.01 | 443 ± 9 |

DOI: https://doi.org/10.7554/eLife.47642.037

above, shear drives the interaction of CD99 with PILR-β1. Thus, this interaction is 'switched-on' under flow, when it is needed and is 'switched-off' once the leukocyte has come to a standstill. This would turn off PILR-β1 support of integrin activation and could tip the balance towards a more prominent contribution of PILR-α. Indeed, we found that PILR-α participates in the extravasation process at a step following leukocyte arrest. PILR-α deficient neutrophils initiated crawling at a lower frequency than WT neutrophils arguing for a role of PILR-α as facilitator of the crawling process. Furthermore, a larger number of WT than PILR-α deficient neutrophils transmigrated per given time period through endothelial monolayers under static conditions and the duration of the diapedesis process per transmigrating neutrophil was prolonged for PILR-α deficient neutrophils in transmigration assays under flow. Thus, the inhibitory PILR-α receptor supports the onset of crawling and enhances transmigration velocity. It is intriguing that a pair of two functionally opposing receptors is able to support the same process, leukocyte extravasation, by acting consecutively at different steps of the extravasation cascade. This sequential action of first PILR-β1 and then PILR-α is made possible by the shear-sensitivity of the PILR-β1/CD99 interaction, which leads to stimulation of PILR-β1 when the cell still moves, whereas this stimulus ceases when the leukocyte comes to a halt, which then allows PILR-α to prevail over PILR-β1 signaling and restrain integrin activation.

This concerted and sequential action of first PILR-β and then PILR-α argues for the benefit of an optimal balance of integrin activation during the extravasation process: a rapid activation of integrins to an optimal level is needed to ensure proper arrest of leukocytes. This then is followed by cell migration, which requires mechanisms that can restrain integrin activation, in order to facilitate proper functioning of integrins during the crawling and transmigration process. Indeed, a defect in integrin inactivation by mutating the cytoplasmic tail of the $\alpha_L$-chain of LFA-1, rendering the integrin constitutively active, leads to enhanced adhesion of leukocytes to the vessel wall accompanied by a defect in transmigration (*Semmrich et al., 2005*). We believe, that the shear-sensitivity of the interaction of PILR-β1 with CD99 regulates the sequential action of both receptors and allows optimal balancing of integrin activation during the leukocyte extravasation process.

Despite the much higher affinity of PILR-α for CD99, it is not clear whether PILR-α on neutrophils is indeed accessible for endothelial CD99 in trans. It was reported that PILR-α is masked by sialylated carbohydrates presented by glycosylated proteins in cis on the surface of neutrophils (*Wang et al., 2013*). Indeed, we confirmed that simply incubating CHO cells expressing PILR-α with a CD99-Fc fusion construct did not yield signals by FACS analysis, while binding was observed after treating the cells with sialidase (*Figure 5—figure supplement 2*). However, using a biochemical approach we could transfer a tag by photoactivation from a CD99-Fc fusion protein to PILR on the cell surface of neutrophils (*Goswami et al., 2017*). The tag was mainly found on the smaller MW form of the two broad, overlapping PILR-bands, which was probably representing PILR-β. This is in line with PILR-α being masked in cis, which suggests that the functional effects we have observed for PILR-α depend on its intrinsic activity (possibly triggered by cis ligands). This is in agreement with our concept of a sequential action of PILR-β1 and PILR-α during the extravasation process, and argues for a balance between CD99-driven PILR-β1 signaling and the intrinsic signaling of PILR-α. Shear driven stimulation of PILR-β1 by endothelial CD99 would tilt the balance towards activation of integrins, whereas static cell attachment would cease the stimulation of PILR-β1 which would favor intrinsic inhibitory signaling of PILR-α supporting deactivation of integrins during neutrophil migration. An interesting aspect

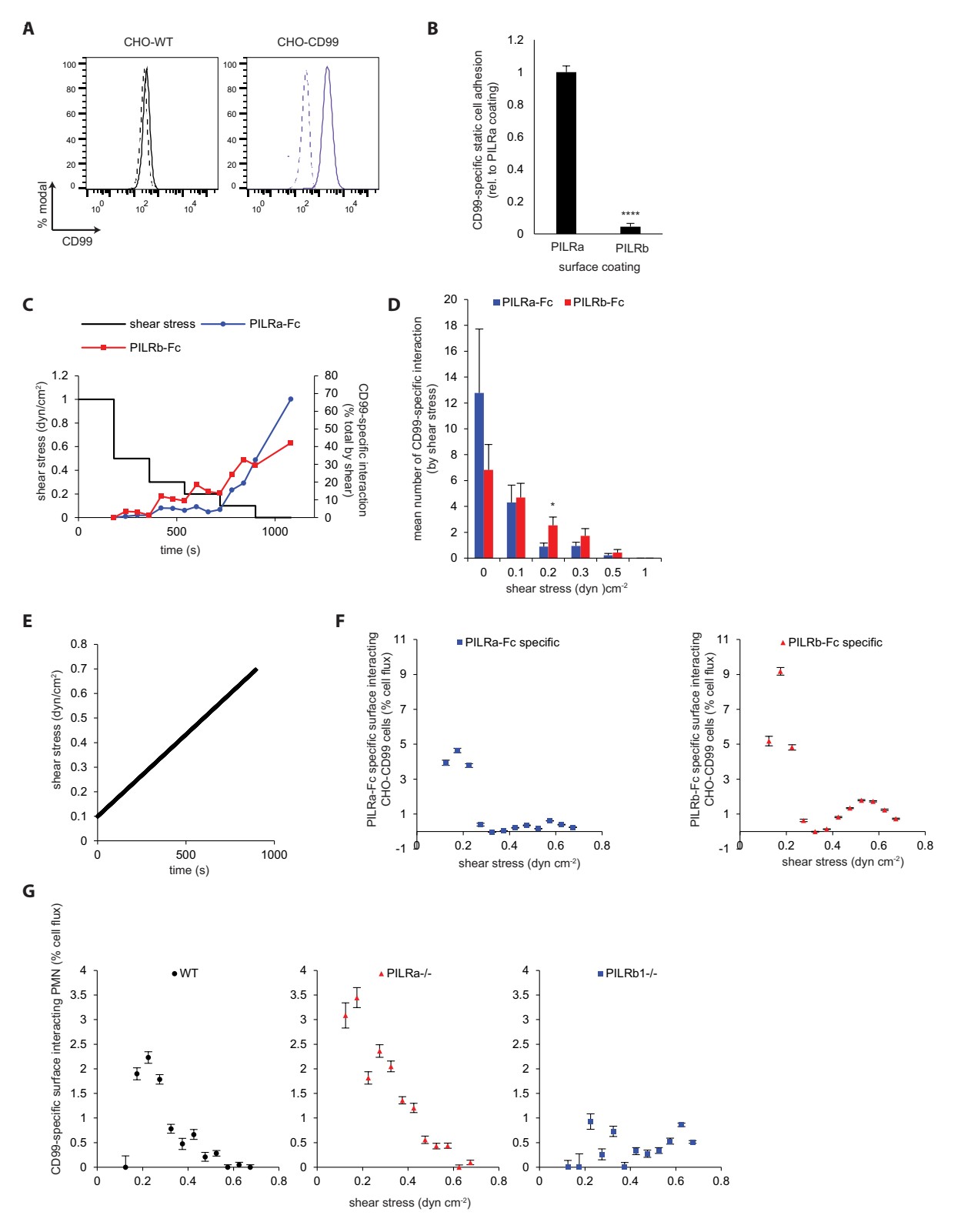

**Figure 5.** Shear stress enhances the interaction of CD99 with PILR-β. (**A**) CHO cells overexpressing CD99 (CHO-CD99) were constructed and analyzed for surface expression of CD99. Solid tracks, anti-CD99; dotted tracks, secondary antibody only. (**B**) 1:1 Cell mixtures of CHO and CHO-CD99 were allowed to adhere onto PILR-α-Fc or PILR-β-Fc coated surface at RT for 10 min (static conditions). CD99-specific cell adhesion was quantified. n = 30 fields from six experiments. (**C**) 1:1 Cell mixtures of CHO and CHO-CD99 were passed over flow chamber surfaces coated with PILR-α-Fc (blue) or PILR-
*Figure 5 continued on next page*

*Figure 5 continued*

β-Fc (red) at RT under stepped shear stress of 3 min interval each (black) and video recorded. CD99-specific cell-surface interactions (≥2 s) were counted every minute. (D) Mean number of CD99-specific interactions under the indicated shear stress (from C) were quantified. n = 8 experiments on PILR-α-Fc coating, n = 9 experiments on PILR-β-Fc coating. (E, F) CHO-CD99 were passed over a surface coated with PILR-α-Fc (blue), PILR-β-Fc (red) or an uncoated control surface under increasing shear stress and video recorded. Transiently interacting (≥50 ms) cells were counted and averaged from six experiments for determining fraction of PILR-specific interacting cell flux. (G) Experiment described as in (E, F) was repeated by passing WT (black), PILR-α$^{-/-}$ (red) and PILR-β1$^{-/-}$ (blue) PMNs through CD99-Fc or control hIgG coated flow chambers. Transiently interacting (≥30 ms) neutrophils were counted and averaged from four experiments for determining the fraction of CD99-specific cell flux interactions. n = 75 measurements per 0.05 dyn/cm$^2$ interval for (E–G). Groups were compared by Mann-Whitney U-test in (B) and 2-tailed t-test in (D). Error bars, SEM. *p<0.05, ***p<.005, ****p<0.001.

DOI: https://doi.org/10.7554/eLife.47642.038

The following source data and figure supplements are available for figure 5:

**Source data 1.** Source data for *Figure 5B*.
DOI: https://doi.org/10.7554/eLife.47642.043
**Source data 2.** Source data for *Figure 5C-D*.
DOI:
**Source data 3.** Source data for *Figure 5E-F*.
DOI: https://doi.org/10.7554/eLife.47642.045
**Source data 4.** Source data for *Figure 5G*.
DOI: https://doi.org/10.7554/eLife.47642.046
**Figure supplement 1.** CD99-specific interaction with coated PILR-Fc under constant flow.
DOI: https://doi.org/10.7554/eLife.47642.039
**Figure supplement 1—source data 1.** Source data for *Figure 5—figure supplement 1*.
DOI: https://doi.org/10.7554/eLife.47642.040
**Figure supplement 2.** CD99-Fc and full-length CD99 produced in CHO cells are glycosylated.
DOI: https://doi.org/10.7554/eLife.47642.041
**Figure supplement 2—source data 1.** Source data for *Figure 5—figure supplement 2A-B*.
DOI: https://doi.org/10.7554/eLife.47642.042

of cis-masking of PILR-α is that PILR-β1 is probably spared from this effect due to its very low affinity under static conditions. This would ensure that PILR-β1 is accessible for CD99 as trans ligand under conditions of flow.

In addition to modulating integrin activation on neutrophils, the PILRs also balance the production of pro- and anti-inflammatory cytokines in infection models. PILR-β gene deficiency strongly reduced pro-inflammatory cytokines and enhanced the levels of anti-inflammatory cytokines in infection models (*Banerjee et al., 2010*; *Tato et al., 2012*). Gene inactivation of PILR-α generally aggravates inflammation, although the mechanisms may vary according to the stimulus and the tissue analyzed. It was reported that the absence of PILR-α augments β2 integrin activation on myeloid cells, which was accompanied with enhanced accumulation of such cells in a chemically induced peritonitis model, and increased monocyte levels in adipose tissue (*Wang et al., 2008*; *Kohyama et al., 2016*). In an autoimmune arthritis model, inflammation was strongly enhanced in the absence of PILR-α, yet this was caused through elevated pro-inflammatory cytokine levels and not by enhanced myeloid cell infiltration (*Sun et al., 2014*). Likewise, PILR-α deficiency did not enhance infiltration of myeloid cells into the *S. aureus* infected peritoneal cavity (*Sun et al., 2014*). All these studies agree on that PILR-α and PILR-β are important balancers of inflammatory responses, with PILR-α acting to restrain inflammation. Balancing the systemic inflammatory cytokine profile is certainly one important mechanism to achieve this.

Depending on the inflammatory model analyzed, PILR-α exerts varying effects on neutrophil extravasation. For example, PILR-α deficiency increased neutrophil extravasation in thioglycollate-induced peritonitis (*Wang et al., 2013*), whereas no such effect was found for the *S. aureus* infected peritoneal cavity (*Sun et al., 2014*). Likewise, in our TNF-α inflamed cremaster model PILR-α deficiency did not lead to enhanced extravasation. The reason for these contrasting results in different inflammation models are probably due to the different cytokine repertoires. It is well known that thioglycollate stimulates the generation of a rich and complex mixture of cytokines in the peritoneum, which differs from that generated upon *S. aureus* infection, and likely from that generated in our

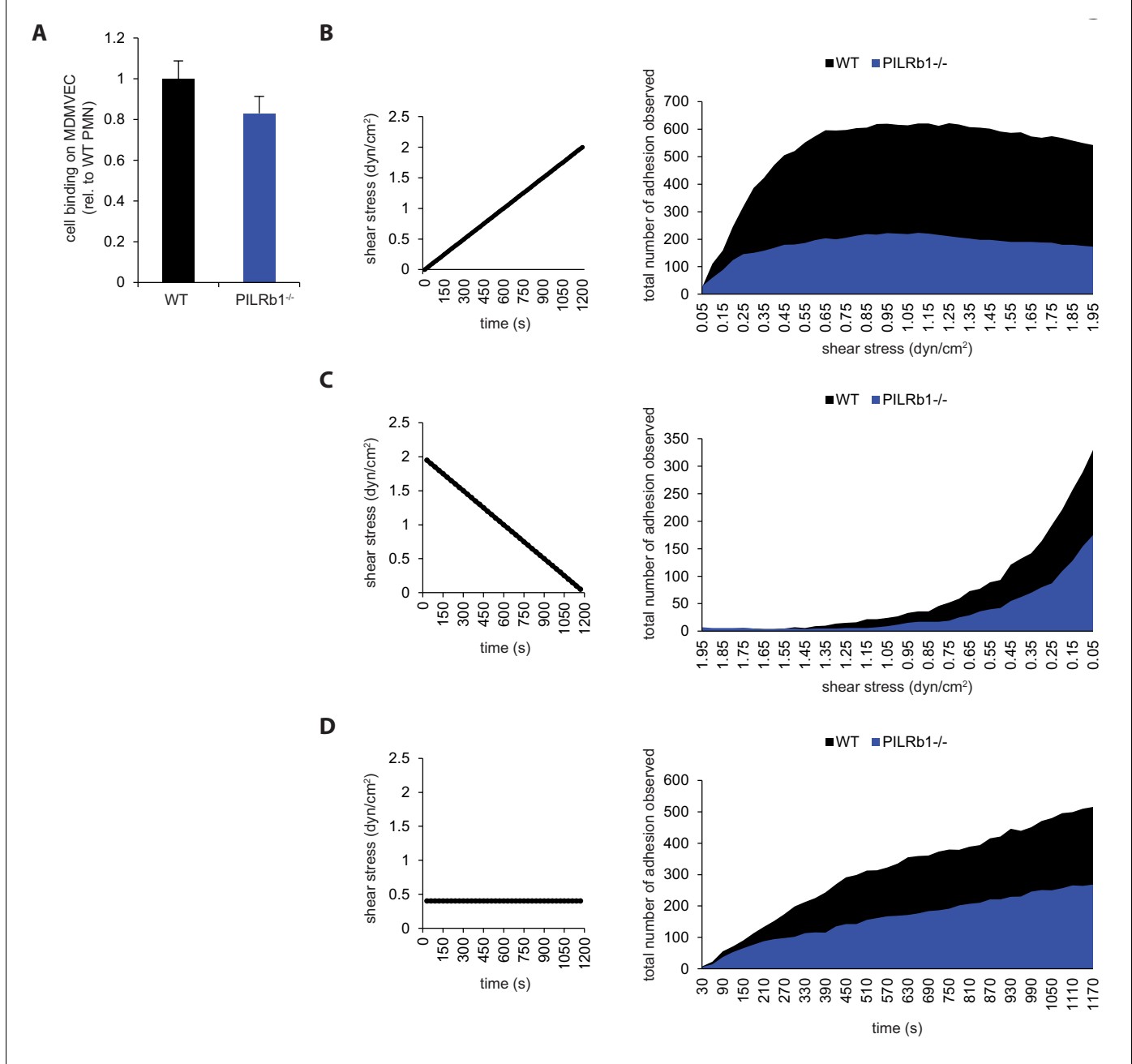

**Figure 6.** Shear switches on PILR-β1-stimulated adhesion of neutrophils to endothelial monolayers under flow. (A) PMN from WT or PILR-β1[-/-] mice were allowed to adhere onto TNF-α inflamed and CXCL-1 pretreated MDMVEC under stasis. Adherent cells were counted. n = 17 fields per genotype from two experiments. Groups were compared by 2-tailed t-test. Error bars, SEM. (B–D) PMN from WT or PILR-β1[-/-] mice were passed under increasing flow over TNF-α inflamed and CXCL-1 pretreated MDMVEC monolayer under an increasing shear gradient (B) or a shear gradient from 2 to 0 dyn/cm[2] (C) or a constant shear at 0.4 dyn/cm[2] (D) in 20 min with adhesion events (C) being pooled from five experiments (B, C) or four experiments (D) for each genotype under different shear stresses.

DOI: https://doi.org/10.7554/eLife.47642.047

The following source data is available for figure 6:

**Source data 1.** Source data for *Figure 6A*.
DOI: https://doi.org/10.7554/eLife.47642.048
**Source data 2.** Source data for *Figure 6B*.
DOI: https://doi.org/10.7554/eLife.47642.049
**Source data 3.** Source data for *Figure 6C*.

*Figure 6 continued on next page*

*Figure 6 continued*

DOI: https://doi.org/10.7554/eLife.47642.050

**Source data 4.** Source data for *Figure 6D*.

DOI: https://doi.org/10.7554/eLife.47642.051

TNF-α inflamed cremaster model. We speculate that a strong and mixed cytokine stimulation, as is induced by thioglycollate, may override the role of PILR-α in crawling and leads to more overall extravasation. With respect to the effect of PILR-α on leukocyte migration it was reported that PILR-α deficiency enhances neutrophil transmigration through ICAM-1-Fc coated transwell filters (*Wang et al., 2013*). We could confirm this effect (*Figure 7—figure supplement 2*). However, such 'transmigration' assays lack endothelial cells. In proper transmigration assays across a monolayer of endothelial cells we found that PILR-α deficiency inhibited this process, pointing towards a function for PILR-α in the transendothelial diapedesis process, which goes beyond simple migration on ICAM-1. Despite this function, we found no net effect on extravasation in the TNF-α inflamed cremaster model in PILR-α deficient mice. We believe the reason for this is that the crawling/diapedesis defect in PILR-α$^{-/-}$ neutrophils is balanced by an increased adhesion.

In conclusion, we show here that PILR-β1 on neutrophils and CD99 on endothelium represent a new pair of important players in the multistep cascade of leukocyte extravasation, which are needed to support the combined function of selectins, chemokines and integrins. Flow induced shear forces support and initiate the interaction between this ligand-receptor pair, which augments chemokine-induced integrin activity and leads to leukocyte arrest. Upon attachment, shear dependent ligand binding is terminated, which via PILR-α leads to integrin deactivation, which may support recycling as it is needed for cell migration. We propose that PILR-β1 and PILR-α represent a receptor pair that is important on neutrophils for the balance of integrin activation during leukocyte extravasation.

# Materials and methods

## Key resources table

| Reagent type (species) or resource | Designation | Source or reference | Identifiers | Additional information |
|---|---|---|---|---|
| Genetic reagent (*M. musculus*) | CD99$^{-/-}$ | PMID: 28223280 | | |
| Genetic reagent (*M. musculus*) | PILR-α$^{-/-}$ | This paper | | |
| Genetic reagent (*M. musculus*) | PILR-β1$^{-/-}$ | This paper | | |
| Cell line (*C. griseus*) | CHO-CD99 | PMID: 15280198 | | |
| Antibody | Rabbit anti-PILR-α C-terminus (VD67, polyclonal) | This paper | | FC: 5 µg/ml |
| Antibody | Rabbit anti-PILRs (VD65, crossreactive, polyclonal) | This paper | | FC: 2 µg/ml Functional assay: 20 µg/ml |
| Antibody | Rat anti-F4/80-FITC (BM8) | Biolegend | Cat# 123107 | FC: 1:100 |
| Antibody | Rat anti-LFA-1-PE (H155-78) | Biolegend | Cat# 141005 | FC: 1:100 |
| Antibody | Rat anti-Ly6G- (APC or FITC) (1A8) | Biolegend | Cat# 127613 Cat# 127605 | FC: 1:250 |
| Antibody | Rat anti-CD11b-PE (M1/70) | Biolegend | Cat# 101207 | FC: 1:100 |
| Antibody | Armenian Hamster anti- CD11c-PE (N418) | Biolegend | Cat# 117307 | FC: 1:100 |

*Continued on next page*

*Continued*

| Reagent type (species) or resource | Designation | Source or reference | Identifiers | Additional information |
|---|---|---|---|---|
| Antibody | Rat anti-CD45-APC (30F11) | Biolegend | Cat# 103111 | FC: 1:100 |
| Antibody | Rat anti-CD182-FITC (SA045E1) | Biolegend | Cat# 149607 | FC: 1:100 |
| Antibody | Mouse anti-NK1.1-PE (PK136) | Biolegend | Cat# 108707 | FC: 1:100 |
| Antibody | Rat anti-B220-PE (RA3-6B2) | BD Biosciences | Cat# 93992 | FC: 1:100 |
| Antibody | Rat anti-CD4-PE (RM4-5) | BD Biosciences | Cat# 26589 | FC: 1:100 |
| Antibody | Rat anti-CD162-PE (2PH1) | BD Pharmingen | Cat# 555306 | FC: 1:100 |
| Antibody | Rat anti-CD8-FITC (53–6.7) | eBioscience | Cat# 11-0081-85 | FC: 1:100 |
| Antibody | Rabbit anti-Syk (polyclonal) | Thermo Fisher | Cat# PA5-27262 | IP: 3 µg/ $5 \times 10^6$ PMN WB: 1:1000 |
| Antibody | Mouse anti-phosphotyrosine (4G10) | Merck Millipore | Cat# 05–321 | WB: 1 µg/ml |
| Peptide, recombinant protein | CD99-Fc | PMID: 15280198 | | |
| Peptide, recombinant protein | P-sel-Fc | PMID: 28223280 | | |
| Peptide, recombinant protein | ICAM-1-Fc | PMID: 28223280 | | |
| Peptide, recombinant protein | TNF-α | PeproTech | Cat# 315-01A | |
| Commercial assay or kit | GeneArt Precision gRNA Synthesis Kit | Thermo Fisher | A29377 | |
| Software, algorithm | FlowJo v10 | FlowJo, LLC | | |
| Software, algorithm | CytExpert | Beckman Coulter | | v2.3.0.84 FACS data acquisition |
| Software, algorithm | Zeiss ZEN (blue edition) | Zeiss | | Image/video acquisition |
| Software, algorithm | Fiji Image J | PMID: 27713081 | | Plugin: Trackmate |

## Mice

PILR-α$^{-/-}$ mice were generated with a 5'-GTCTGACCAGACCCGGTACTTTGG-3' (PAM underlined) guide sequence, which targeted exon-2 of the pilra locus and induced frame-shift indels. Plasmid pX330 encoding the gRNA and Cas9 was microinjected into the pro-nucleus of fertilized mouse oocytes. 432 injected oocytes gave rise to 41 pups of which three independent mouse lines were established lacking PILR-α protein. Potential mutations at off-target sites were excluded by sequencing: NM_133209, NM_001024932, NM_001198971, NM_001163557, NM_053271, NM_009675, NM_010099, NM_028833. For genotyping the WT pilra allele, 5'-GACCAGACCCGGTACTT-3' and 5'-TACCCATGGATTTCTAGCAG-3' were used. For mutant pilra allele, a common sense primer 5'-ATGGCTTTGCTGATCTCG-3'was used with line specific antisense primers: 5'-GAAAAACTCTGCCGGG-3', 5'-CTCTGCCAAAG TCTCC-3', 5'-TTGCAGAAAAACTCCG-3'.

The PILR-β locus (harboring the 92% homologous, adjacent genes pilrb1 and pilrb2) was targeted with a pair of gRNAs with guide sequences of 5'-CCCCACGTGGATTGCATATCAGG-3' targeting intron-2 of both pilrb1 and pilrb2 loci and 5'-AAGCATTTCGCGTGGGTAGAAGG-3' targeting intron-4 of pilrb1 locus. gRNAs were prepared with GeneArt Precision gRNA Synthesis Kit (Thermo Fisher, A29377). RNA mixture containing 20 ng/µl Cas9 mRNA (Thermo Fisher, A29378) and 40 ng/µl of each gRNA was microinjected into the cytoplasm of fertilized mouse oocytes. Of 147 pups 38 carried mutations at the PILR-β locus. four equivalent founder lines lacking pilrb1 transcription were established with a deletion of a 3.1 kDa between intron-2 and intron-4 of the pilrb1 locus. Potential mutations at off-target sites were excluded by sequencing: NM_001081176, NM_008886, NM_009547, NM_011547, NM_021423, NM_024243, NM_153510. For genotyping WT pilrb1, 5'-AC

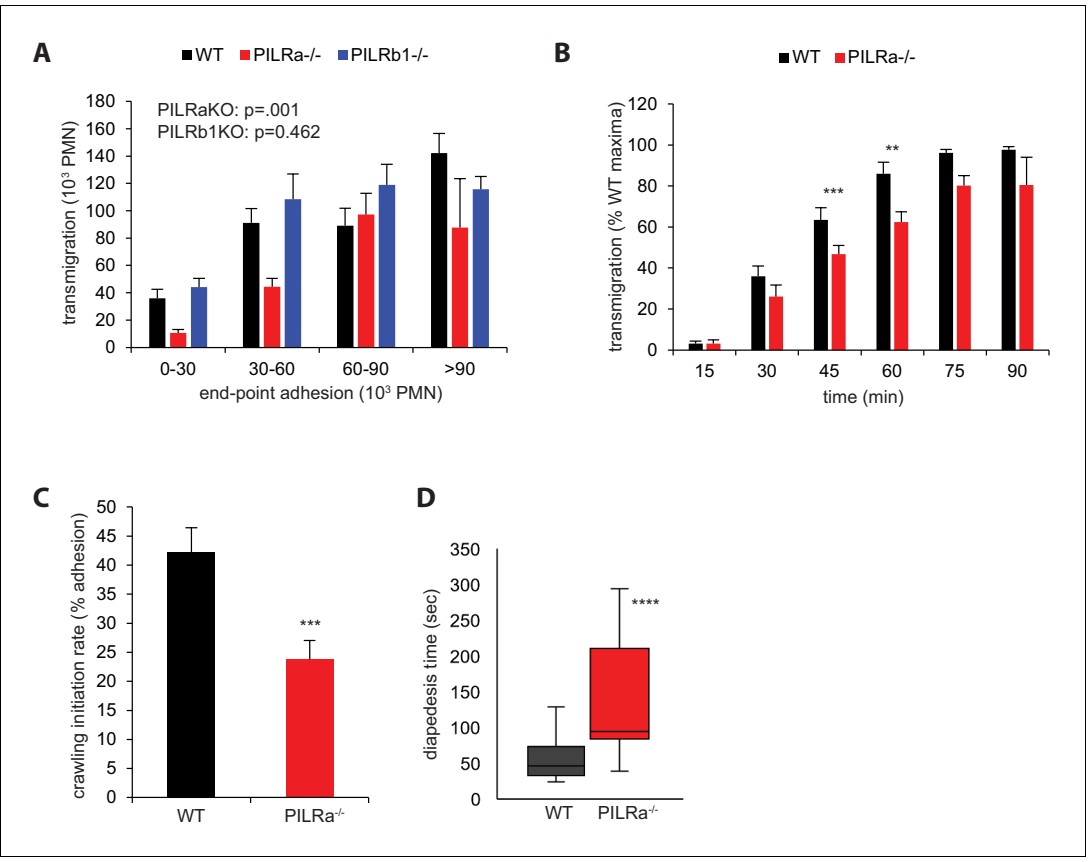

**Figure 7.** PILR-α supports crawling and diapedesis. (**A**) Correlation between adhesion and transmigration of WT, PILR-α⁻/⁻ and PILR-β1⁻/⁻ PMNs in static transendothelial cell migration (TEM) assays through MDMVEC. n = 96 (WT), 54 (PILR-α⁻/⁻) and 41 (PILRβ1⁻/⁻) transwells. Knockouts were compared against WT by 2-way ANOVA followed by Holm-Sidak method for multiple comparisons. (**B**) WT and PILR-α⁻/⁻ PMN were allowed to transmigrate through MDMVEC for the indicated times. n = 4 transwells per time point per genotype from two experiments. Genotypes were compared by 2-tailed paired t-test. (**C, D**) WT and PILR-α⁻/⁻ PMNs with either genotype fluorescently labeled were mixed and passed over TNF-α inflamed MDMVEC monolayers to allow transendothelial migration at 37°C for 20 min under flow, analyzed by video recording. The fraction of crawling cells (**C**) and the time needed for diapedesis (**D**) were determined. n = 14 videos for (**C**) and n = 31 (WT) or 15 (PILR-α⁻/⁻) diapedesis events for (**D**). Groups were compared by Mann-Whitney U-test. Error bars: SEM. **p<0.01, ***p<.005, ****p<0.001.

DOI: https://doi.org/10.7554/eLife.47642.052

The following source data and figure supplements are available for figure 7:

**Source data 1.** Source data for *Figure 7A*.
DOI: https://doi.org/10.7554/eLife.47642.059
**Source data 2.** Source data for *Figure 7B*.
DOI: https://doi.org/10.7554/eLife.47642.060
**Source data 3.** Source data for *Figure 7C-D*.
DOI: https://doi.org/10.7554/eLife.47642.061
**Figure supplement 1.** PILR-α suppresses adhesion on endothelial monolayers.
DOI: https://doi.org/10.7554/eLife.47642.053
**Figure supplement 1—source data 1.** Source data for *Figure 7—figure supplement 1A*.
DOI: https://doi.org/10.7554/eLife.47642.054
**Figure supplement 1—source data 2.** Source data for *Figure 7—figure supplement 1B*.
DOI: https://doi.org/10.7554/eLife.47642.055
**Figure supplement 2.** Effect of PILR-α on leukocyte crawling requires endothelial surface.
DOI: https://doi.org/10.7554/eLife.47642.057
**Figure supplement 2—source data 1.** Source data for *Figure 7—figure supplement 2*.
DOI: https://doi.org/10.7554/eLife.47642.058

*Figure 7 continued*

**Figure supplement 3.** PILR-α on cellular surface is unavailable for CD99-Fc binding due to blockade by cis-interaction with sialylated entities.

DOI: https://doi.org/10.7554/eLife.47642.056

TTTGAGTT-GGGCATGTGTAA-3' and 5'-TCCTTCTACCCACGCGA-3' were used. For mutant pilrb1, 5'-GCAACTGAAGTCCCCTAGACT-3' and 5'-CTGAGACGTAGAGGAC-AATCG-3' were used. Major in vitro and in vivo phenotypes (*Figures 1*, *2*, *6* and *7*) were reproduced with at least two PILR-α[-/-] and PILR-β1[-/-] lines.

Gene-inactivated mice were backcrossed ≥5 times to the C57Bl/6 background. Animals were maintained in a barrier facility under special pathogen-free conditions. All animal experiments were carried out under German legislation for the protection of animals and approved by the Landesamt für Natur Umwelt und Verbraucherschutz Nordrhein-Westfalen under the reference number AZ 84–02.04.2017.A101.

## Antibodies against PILRs

Polyclonal rabbit antiserum VD65 crossreactive for PILR-α and PILR-β was generated against mouse PILR-β-Fc-fusion protein as described (*Goswami et al., 2017*). For affinity purification, antibodies were depleted against human IgG1, then affinity purified against the immunogen. To enrich for PILR-β specific antibodies, crossreactive anti-PILR-α antibodies were depleted against PILR-α-Fc and residual antibodies were affinity purified against PILR-β-Fc. Rabbit antiserum VD67 specific for PILR-α was generated against the C-terminal peptide GNPQEETVYSIVKAK.

## Cell lines

CHO-dhfr[-] (denoted as CHO in the study) is originally from ATCC and cultured in our laboratory. CHO cells were used as expression platforms in our study. The expressed protein(s), for each line, are confirmed by flow cytometry with specific antibodies. Cell lines are routinely tested for myco-plasma contamination in our laboratory.

## Flow cytometry

For surface antigen analysis, bone marrow cells were gated for Ly6G[+] neutrophils; for subset characterization, cells were gated for CD45[+] leukocytes. $2 \times 10^4$ target cells were analyzed for indicated surface antigens. For PILR-α detection, C-terminus-specific pAb VD67 was used for staining

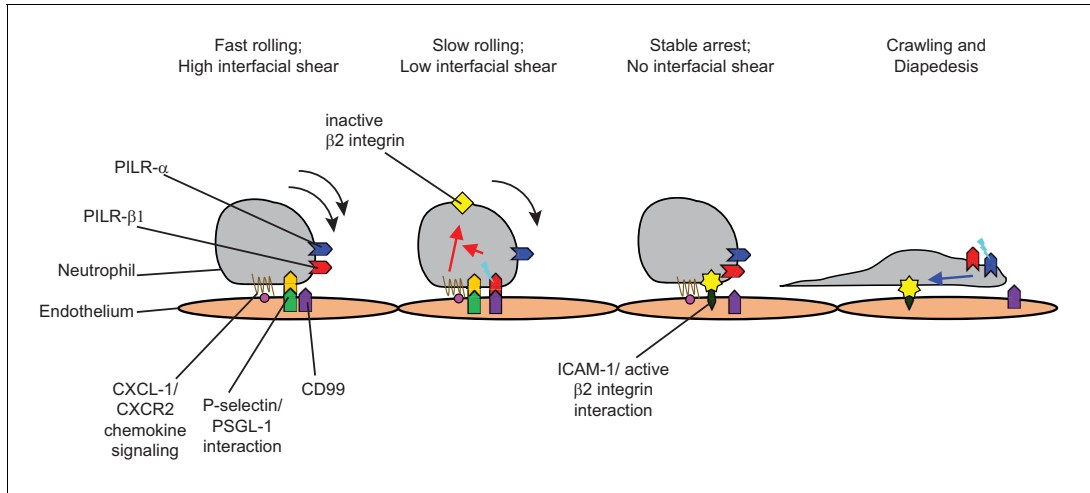

**Figure 8.** Graphical depiction. An antagonistic pair of receptors, PILR-β1 and PILR-α, supports neutrophil arrest and seamless transition to crawling via modulating β2 integrin activity at different steps during extravasation.

DOI: https://doi.org/10.7554/eLife.47642.062

permeabilized cells, followed by secondary antibody detection. Expression levels were determined by mean fluorescence intensity (MFI) normalized to WT. The following antibodies were used: Anti-PILR serum VD65, crossreactive for extracellular domains of both PILR subtypes (*Goswami et al., 2017*); anti-F4/80-FITC (BM8), anti-LFA-1-PE (H155-78), anti-Ly6G-(APC or FITC) (1A8), anti-CD11b-PE (M1/70), anti-CD11c-PE (N418), anti-CD45-APC (30F11), anti-CD182-FITC (SA045E1), anti-NK1.1-PE (PK136) all from Biolegend; anti-B220-PE (RA3-6B2), anti-CD4-PE (RM4-5), anti-CD162-PE (2PH1) all from BD Biosciences; and anti-CD8-FITC (53–6.7, eBioscience).

## Quantitative RT-PCR

cDNA was prepared by reverse transcription of RNA extracted from peripheral blood leukocytes with oligo(dT)12–18 (Thermo Fisher, 18418012) for first-strand synthesis. Expression levels of the indicated genes were detected by using TaqMan probes (Thermo Fisher): pilra (Mm04211819_m1), pilrb1 (Mm00652421_m1), pilrb2 (Mm04214229_s1); gapdh (Mm99999915_g1) was used as internal control.

## Intravital microscopy

12–17 week old male mice received intrascrotal TNF-α stimulation for 2 hr, and cremaster muscles were analyzed by IVM as described (*Bixel et al., 2007*; *Broermann et al., 2011*). Vessels of 20–40 μm in diameter were observed. In some experiments, bone marrow transplantation was performed as described (*Goswami et al., 2017*) followed by IVM 6 weeks later.

## Transendothelial migration assay

To profile the correlation between neutrophil adhesion and transendothelial migration (TEM), a standard TEM assay (*Artz et al., 2016*) was performed with modifications. Primary mouse dermal endothelial cells (MDMVEC) were isolated and cultured as described (*Frye et al., 2015*). PMNs were labeled with 2 μM CMFDA at 37°C for 30 min, and resuspended at densities between 1 to $15 \times 10^6$ per ml in assay medium to achieve different levels of adhesion and corresponding transmigration. 40 ng/ml CXCL-1 was administered per lower chamber and 100 μl of neutrophil-suspension was added per well followed by 40 min transmigration at 37°C. Filters were washed twice, lysed in 150 ul 2% SDS and lysates were measured for excitation/emission at 485/528 nm with a fluorescence reader and calibrated against a standard curve to estimate PMN adhesion. Transmigrated cells were counted by a CASY cell counter (OLS). Adhesion and transmigration measurements were recorded in pair for every transwell. Data were categorized by adhesion, and transmigration of indicated genotypes was analyzed by 2-way ANOVA (SigmaPlot). Time course experiments of TEM assays were performed similarly, except for a fixed input of $5 \times 10^5$ in 100 μl assay medium and termination of transmigration at indicated time points.

## Real time monitoring of neutrophil interactions with MDMVEC under flow

three $\times 10^4$ MDMVEC were seeded into gelatin-coatead flow chamber (as above). Confluent cells were stimulated with 5 nM TNF-α (PeproTech) for at least 16 hr. WT or PILR-β1$^{-/-}$ neutrophils were labeled with 2 μM CMFDA, resuspended at $5 \times 10^5$ per ml MC-buffer, passed over the monolayer at 0 to 2 dyn/cm$^2$ in 20 min, and video-recorded at 0.33 fps under a 10x objective. Adhesion traces were analyzed by TrackMate (*Tinevez et al., 2017*).

To observe in vitro transmigration under flow, MDMVEC were seeded into laminin-coated flow chambers (as above) and one day later stimulated with TNF-α as above. Either genotype of WT or PILR-α$^{-/-}$ neutrophils were labeled with CMFDA, mixed 1:1, at a density as above, passed through the flow chamber at 1 dyn/cm$^2$ for 20 min, and video-recorded at 0.33 fps under a 10x objective. Crawling was monitored by TrackMate (*Tinevez et al., 2017*).

## Soluble ICAM-1-Fc binding assay

Secondary antibody complexed ICAM-1-Fc detection mixture was prepared by incubating 0.5 μg ICAM-1-Fc with 1.5 μl Donkey anti-Human IgG-PE (Jackson Laboratory, 709-116-149), 0.2 μl anti Ly6G-APC (1A8 Biolegend) in 3.5 μl at room temperature (RT) for 15 min in the dark. Crosslinking antibody (purified rabbit anti-PILRs (VD65) or control pre-serum) was diluted to 60 μg/ml in HBSS

containing 10 mM HEPES buffered at pH7.0 resuspension buffer (RB). Bone marrow cells were prepared in $2 \times 10^8$ per ml RB, rested for 30 min and blocked for FcR with 2.4G2. Reaction mixture was prepared by mixing 5 µl cells, 5 µl crosslinking antibody and 3.5 µl detection mixture at RT for 15 min. Reactions were initiated by adding 1.5 µl CXCL-1 and Mg/CaCl$_2$ to a final concentration of 100 ng/ml and 1 mM respectively. Aliquots were fixed at indicated time points with equal volume of 4% PFA, washed and analyzed by FACS.

## Adhesion assays

CD99-induced adhesion of neutrophils to ICAM-1 under flow was tested in flow chambers (µ-Slide VI$^{0.4}$, uncoated, ibidi) coated with 1 µg/ml CXCL-1, 5 µg/ml P-selectin-Fc, 5 µg/ml ICAM-1-Fc, and 10 µg/ml CD99-Fc or hIgG in PBS at 4°C overnight and blocked with 1% casein (Thermo Fisher) at RT for 2 hr. Proper modification of CD99-Fc with sialic acid was tested by neuraminidase (from *Arthrobacter ureafaciens*, Roche) treatment (*Figure 5—figure supplement 2*). To allow competition between WT and PILR-$\alpha^{-/-}$ or PILR-$\beta1^{-/-}$ neutrophils, before Fc receptor blockade, WT neutrophils were labeled with 2 µM Cell Tracker Green (CMFDA) (Thermo Fisher, C7025) in RB at 37°C for 30 min. WT neutrophils were mixed 1:1 with PILR-$\alpha^{-/-}$ or PILR$\beta1^{-/-}$ neutrophils, and diluted to $5 \times 10^5$ per ml each with MC-buffer. Assays were performed at 5 dyn/cm$^2$.

To compare CHO-CD99 binding to PILR-$\alpha$ and PILR-$\beta$ under static conditions, 10 ug/ml PILR-$\alpha$-Fc or PILR-$\beta$-Fc was coated at 4°C overnight, followed by casein blockade. CHO-CD99 were labeled with CMFDA (as above) and mixed 1:1 with CHO each at $2.5 \times 10^6$ per ml in MC-buffer and incubated with coated surfaces at RT for 10 min, followed by washing, fixation and counting under a 10x objective. CD99-specific adhesion was defined as the number of adherent CHO-CD99 subtracted by the number of adherent CHO cells. Proper modification of CD99 on CHO cells with sialic acid was tested by neuraminidase treatment of imunoprecipitated CD99 (*Figure 5—figure supplement 2*).

## Immunoprecipitation and Western blot

$5 \times 10^6$ neutrophils were treated with 20 µg/ml crosslinking antibody (purified rabbit anti-PILRs (VD65) or control IgG) in 67.5 µl at RT for 15 min. Cells were stimulated by adding 1.5 µl CXCL-1 and Mg/CaCl$_2$ to a final concentration of 100 ng/ml and 1 mM, respectively. Unstimulated samples and samples stimulated for 30, 60 or 75 s were fixed in 2% PFA at RT for 5 min, washed and lysed in 400 ul lysis buffer (1% TritonX-100, 0.1% sodium deoxycholate, 140 mM NaCl, 1 mM EDTA, 10 mM Tris-HCl pH7.5, protease inhibitor cocktail (Roche), 300 uM orthovanadate, 1 mM NaF). Lysate was immunoprecipitated with 3 µg anti-Syk (Thermo Fisher, PA5-27262) or rabbit IgG on Protein G dynabead (Thermo Fisher) at 4°C overnight. Immunoprecipitated samples were analyzed by SDS-PAGE and blotted for phospho-tyrosine on Syk (4G10) and total Syk protein (Thermo Fisher, PA5-27262).

## Monitoring CHO-CD99 interactions with PILR-coated surfaces under flow

Flow chambers (as above) were coated with 50 µg/ml PILR-$\alpha$-Fc or PILR-$\beta$-Fc in PBS at 4°C overnight, followed by casein blockade. CHO cells were labeled (as neutrophils above), mixed 1:1 with CHO-CD99 at $7.5 \times 10^5$ per ml each in MC-buffer and passed through the flow chamber at a step-down shear stress gradient of 1, 0.5, 0.3, 0.2, 0.1, 0 dyn/cm$^2$, in 3 min intervals at RT. Flow at 1 dyn/cm$^2$ was applied to clear the observation field after the static interval. Video-recording was at 0.5 frame per second (fps) under a 10x objective. Interactions were defined as cells stopping under flow for at least one frame. CD99-specific interactions were defined as number of interacting CHO-CD99 minus the number of interacting CHO cells for a particular frame. From 180 s onwards, cell-surface interactions were counted every minute except for the static interval, where interactions were counted immediately after the field-clearing laminar flow pulse.

In other experiments, CHO-CD99 were resuspended at $3.75 \times 10^5$ per ml in MC-buffer, passed through flow chambers either coated with PILR-$\alpha$-Fc or PILR-$\beta$-Fc or uncoated under a ramping shear from 0.1 to 0.7 dyn/cm$^2$ in 15 min at RT, and videos recorded at 1 fps and 50 ms brightfield exposure. Surface interacting cells were identified at each frame by TrackMate (*Tinevez et al., 2017*) and normalized to input cells. PILR-specific interaction was defined as the interaction flux on either PILR-coated surface subtracted by the interaction flux on uncoated surfaces, and was determined for

each frame. Average interaction fluxes were plotted against the corresponding mean shear stress at 0.05 dyn/cm$^2$ intervals.

## Statistics

2-tailed Student's t-test or Mann-Whitney U-test was used to compare two groups. One-sample t-test was used to compare a control-normalized group against a value. To compare more than two groups, 1-way ANOVA, 2-way ANOVA or ANOVA on rank was used followed by a post-test indicated for each experiment. A difference was considered as statistically significant for an α-value of 0.05.

## Acknowledgements

We thank Tanja Möller fpr her excellent technical assistance. This work was supported by funds from the Deutsche Forschungsgemeinschaft (SFB1009, A1 to DV) and by the Excellence Cluster Cells in Motion.

## Additional information

### Funding

| Funder | Grant reference number | Author |
| --- | --- | --- |
| Deutsche Forschungsge-meinschaft | SFB1009 | Debashree Goswami |
| Deutsche Forschungsge-meinschaft | SFB1009, A1 | Debashree Goswami |

The funders had no role in study design, data collection and interpretation, or the decision to submit the work for publication.

### Author contributions

Yu-Tung Li, Conceptualization, Formal analysis, Investigation, Writing—original draft, Writing—review and editing; Debashree Goswami, Melissa Follmer, Annette Artz, Formal analysis, Investigation; Mariana Pacheco-Blanco, Investigation, Methodology; Dietmar Vestweber, Conceptualization, Supervision, Funding acquisition, Writing—original draft, Project administration, Writing—review and editing

### Author ORCIDs

Yu-Tung Li https://orcid.org/0000-0002-0718-7344
Dietmar Vestweber https://orcid.org/0000-0002-3517-732X

### Ethics

Animal experimentation: All animal experiments were carried out under German legislation for the protection of animals and approved by the Landesamt für Natur Umwelt und Verbraucherschutz Nordrhein-Westfalen under the reference number AZ 84-02.04.2017.A101.

### Decision letter and Author response

Decision letter https://doi.org/10.7554/eLife.47642.066
Author response https://doi.org/10.7554/eLife.47642.067

## Additional files

### Supplementary files

• Transparent reporting form
DOI: https://doi.org/10.7554/eLife.47642.063

## Data availability

All data generated or analysed during this study are included in the manuscript and supporting files. Source data files for the figures have been provided.

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
