## [Decision Letter]

Thank you for submitting your article "Blood flow guides sequential support of neutrophil arrest and diapedesis by PILR-β1 and PILR-α" for consideration by *eLife*. Your article has been reviewed by three peer reviewers, including Reinhard Fässler as Reviewing Editor and Reviewer #1, and the evaluation has been overseen by Tadatsugu Taniguchi as the Senior Editor. The following individual involved in review of your submission has agreed to reveal their identity: Francisco Sánchez Madrid (Reviewer #3).

The reviewers have discussed the reviews with one another and the Reviewing Editor has drafted this decision to help you prepare a revised submission.

Summary:

The manuscript shows that (1) PILR-β1 facilitates chemokine effects during integrin-mediate neutrophil adhesion to endothelial CD99 in postcapillary venules, (2) PILR-β1 binding to CD99 is shear force-dependent, (3) PILR-β1crosstalks upstream of the chemokine/GPCR signaling-mediated calcium influx, and (4) PILR-α facilitates neutrophil crawling and transendothelial migration upon PILR-β1 neutrophil arrest.

Essential revisions:

1) The PILR-α data are in conflict with the paper published by Wang et al., 2013. This must be clarified.

2) Figure 6: add experiments under constant low shear stress conditions kept over 20 min (1200s) to exclude that the observed differences depend on the duration of the experiment rather than on changes in shear stress. In addition, one may also run those experiments starting with high shear stress and then reduce shear stress in a step like fashion.

3) The mechanism through which PILR-β1 and PILR-α control adhesion under shear flow, in particular whether they affect the degree of integrin activation should be addressed. The authors should consider whether a transient phosphorylation-dephosphorylation of the tyrosine kinase Syk may be involved in the mechanism by which PILR-α/β1 regulates the on-off switch of the integrin.

[Editors' note: further revisions were requested prior to acceptance, as described below.]

Thank you for resubmitting your work entitled "Blood flow guides sequential support of neutrophil arrest and diapedesis by PILR-β1 and PILR-α" for further consideration at *eLife*. Your revised article has been favorably evaluated by three reviewers, including Reinhard Fässler as the Reviewing Editor, and the evaluation has been overseen by Tadatsugu Taniguchi as the Senior Editor. The manuscript has been improved but there are some remaining issues that need to be addressed before acceptance, as outlined below:

1) Off target effects: according to the Materials and methods section, several independent null lines were generated. Please mention in the text that at least two Crispr/Cas9-induced PILR-α- and PILR-β1-null mutation produced the same in vivo and in vitro phenotypes.

2) Define rolling flux fraction as number of rolling leukocytes expressed as a percent of all leukocytes traveling through microvessel.

3) Label PILR-β^-/-^ throughout the paper as PILR-β1^-/-^ to indicate that the gene inactivation specifically targeted only the PILR-β1 isoform

4) Does the arrest of WT and mutant neutrophils on control IgG differ? Can this be clarified in the manuscript?

5) Syk inhibition experiments should be done in the presence of CD99-Fc. Please repeat this assay accordingly.

6) The discrepant findings between the present manuscript by the Vestweber lab and the published Wang et al. (NI) report are not well handled. Firstly, the authors should show their in vitro transmigration data though ICAM-1-Fc coated transwell filters in their manuscript. Secondly, their arguments regarding the values and pitfalls of the different in vivo extravasation models used in the two studies should be better discussed. They have a strong argument that the 'thioglycollate' trigger may produce a non-physiological response (storm) that overrides effects caused by the PILR-α-deficiency. It would have been wonderful if the authors would have shown the 'thioglycollate' and the 'TNF/cremaster muscle' in their paper side by side. Although this would have been the best argument for their findings, they should at least improve their discussion.

---

## [Author Response]

Essential revisions:1) The PILR-α data are in conflict with the paper published by Wang et al., 2013. This must be clarified.

We are aware that our data on PILR-α differ from the results published by Wang et al., 2013. We will first discuss the in vitro, then the in vivo results.

Our in vitro results are in part in agreement with the published results. We found that adhesion of CXCL-stimulated PILR-α^-/-^ neutrophils to ICAM-1-Fc or to TNF-α-stimulated endothelial cells is increased in comparison to similarly treated WT neutrophils (Figure 7—figure supplement 1). This was also shown by Wang et al. for fMLF-stimulated PMNs, however only analyzed for ICAM-Fc; not for endothelial cells. Our results differ when it comes to transmigration assays:

Importantly, we analyzed transmigration of PMNs through endothelial cell layers grown on transwell filters, Wang et al. analyzed transmigration with ICAM-1-Fc coated transwell filters. We analyzed transmigration for different time periods, and found that fewer PILR-α^-/-^ deleted PMNs transmigrated after 45 and 60 minutes; at later time points the number of transmigrated cells plateaued and PILR-α^-/-^ cells caught up (Figure 7B). In contrast, PILR-α^-/-^ PMNs transmigrated more efficiently than WT PMNS, however, transmigration was only analyzed for ICAM-1-Fc coated filters. We have now repeated such experiments and find a tendency of slightly enhanced transmigration of PILR-α^-/-^ PMNs through ICAM-1-Fc coated filters, although the effect was not significant (p=0.0658).

**Author response image 1. respfig1:** Transmigration of WT and PILR-α^-/-^ PMNs towards 40ng/ml CXCL1 through polycarbonate transwell filters (3μm pore size) coated with 2μg/ml ICAM-1-Fc for 1 h at 37°C. (P value is 0.0658 (2-tailed t-test)).

Thus, this type of transmigration assay (in the absence of endothelium) may reflect the enhanced adhesion of PMNs to ICAM-1. However, a proper transmigration assay with endothelial cells is more complex and reflects the diapedesis process through a cell layer. We suggest that PILR-α supports the diapedesis process in ways, which may go beyond the simple activation of β_2_-integrins on leukocytes. In agreement with these results, we also found that the duration of the diapedesis process was extended 2 fold for PILR-α^-/-^ PMNs.

Collectively, we believe that our results are not in contradiction with those by Wang et al., since the experimental conditions of the transmigration assays differ significantly. In vivo, we found that deletion of PILR-α slightly enhanced adhesion in postcapillary venules of the TNF-α stimulated cremaster muscle, although this effect did not translate into more extravasation, which is in agreement with our in vitro results: enhanced adhesion was compensated by reduced transmigration efficiency. In contrast to our results in the TNF-α-stimulated cremaster muscle, Wang et al. found that thioglycollate stimulation of the peritoneum enhanced recruitment of PILR-α^-/-^ PMNs more efficiently than recruitment of WT PMNs. However, as we pointed out in the Discussion of our manuscript (sixth paragraph), a group at Genentech (Sun et al., 2014) found that infection of the peritoneal cavity with *S. aureus* did *not* enhance the recruitment of PILR-α^-/-^ neutrophils more efficiently than WT neutrophils (analyzed at 2h and 6h post infection). Thus, even in the same tissue (peritoneal cavity), the contribution of PILR-α to PMN recruitment differs, dependent on the type of inflammatory stimulus. Thioglycollate is a complex mixture of chemicals described as “thioglycollate-bouillon” which is left to “mature” for several weeks before use. It is known to stimulate intraperitoneal macrophages to produce a rich cocktail of pro-inflammatory factors and chemoattractants which act in combination to activate endothelium and attract neutrophils. At present, we assume that thioglycollate may trigger some chemoattractant, that contributes to integrin activation, which may be induced much less upon *S. aureus* infection or upon stimulation of the cremaster with TNF-α. If the receptor for this chemoattractant would be more sensitive to the inhibitory action of PILR-α than the chemokine receptor that is mainly responsible for TNF-α-induced or *S.aureus*-induced PMN recruitment, the discrepancy of the results by Wang et al., on one side and the results by Sun et al., and our group on the other side could be explained. We have slightly changed the corresponding paragraph in our Discussion, to better address this point.

In addition to the thioglycollate-peritonitis data, Wang et al., also reported results obtained with a septic shock model based on the systemic stimulation with LPS. Higher MPO activities, which did not distinguish between intravascular and extravascular neutrophils, were detected in lung and liver tissues from PILR-α^-/-^ mice. However, the liver of PILR-α^-/-^ mice showed signs of haemorrhage, clearly indicating vascular damage. Thus, the higher number of PMNs found in liver and lung might not have been a result of a proper multistep extravasation process across an intact endothelial barrier. Thus, this model is very complex and difficult to compare with our results obtained in a well-defined extravasation model which allows direct analysis of the different steps of the process.

2) Figure 6: add experiments under constant low shear stress conditions kept over 20 min (1200s) to exclude that the observed differences depend on the duration of the experiment rather than on changes in shear stress. In addition, one may also run those experiments starting with high shear stress and then reduce shear stress in a step like fashion.

In order to exclude that the observed differences in binding of WT and PILR-β^-/-^ PMNs to cytokine/chemokine treated endothelial cells depend on the duration of the experiment, we have now chosen a constant low shear (0.4dyn/cm^2^) for 20 minutes (new Figure 6C). This way we could show that it is indeed the shear stress which improves PMN binding to cytokine induced endothelial cells in a PILR-β-dependent way.

In addition, we have reversed the experiment starting with high shear and then reduced shear stress (from 2 – 0 dyn/cm^2^) over a period of 20 min (new Figure 6B). Again, we found that shear supported PMN binding to cytokine/chemokine treated endothelial cells in a PILR-β dependent way.

3) The mechanism through which PILR-β1 and PILR-α control adhesion under shear flow, in particular whether they affect the degree of integrin activation should be addressed. The authors should consider whether a transient phosphorylation-dephosphorylation of the tyrosine kinase Syk may be involved in the mechanism by which PILR-α/β regulates the on-off switch of the integrin.

We have now analyzed whether stimulation of PILR modulates chemokine-induced tyrosine phosphorylation of Syk. We found that CXCL1-induced tyrosine phosphorylation of Syk was strongly enhanced if PMNs were incubated with crosslinking polyclonal antibodies recognizing the extracellular domain of both PILR forms (Figure 3—figure supplement 1A). To test the relevance of Syk for the modulating effect of PILR-α and PILR-β1 on the adhesive function of β_2_-integrins, we performed adhesion assays with immobilized ICAM-1-Fc and chemokine-stimulated PMNs of all three genotypes treated with or without the Syk inhibitor PRT-060318. We found that the Syk inhibitor blocked the adhesion modulating effects of each of the two PILR receptors, (i.e. adhesion of the KO cells was as efficient as that of WT cells if all cells were treated with the inhibitor).

[Editors' note: further revisions were requested prior to acceptance, as described below.]

The manuscript has been improved but there are some remaining issues that need to be addressed before acceptance, as outlined below:

*1) Off target effects: according to the Materials and methods section, several independent null lines were generated. Please mention in the text that at least two Crispr/Cas9-induced PILR-α - and PILR*-β1*-null mutation produced the same* in vivo *and* in vitro *phenotypes.*

We have now stated in the Materials and methods section that major in vivo and in vitro experiments were produced with more than one PILR-α- and PILR-β1- null mouse line.

2) Define rolling flux fraction as number of rolling leukocytes expressed as a percent of all leukocytes traveling through microvessel.

Rolling flux fraction has now been defined in the subsection “PILR-α and PILR-β1 exert different functions in neutrophil extravasation”.

*3) Label PILR*-β*^-/-^ throughout the paper as PILR*-β1*^-/-^ to indicate that the gene inactivation specifically targeted only the PILR*-β1*isoform.*

This has now been corrected.

4) Does the arrest of WT and mutant neutrophils on control IgG differ? Can this be clarified in the manuscript?

This has now been stated in the subsection “PILR-β1 is required for CD99 support of chemokine-induced neutrophil binding to ICAM-1”.

5) Syk inhibition experiments should be done in the presence of CD99-Fc. Please repeat this assay accordingly.

The Syk inhibition experiments have now been done in the presence of CD99-Fc and the results are shown in the new parts G and H of Figure 3 and are described in the Results subsection “PILR-β1 is required for CD99 support of chemokine-induced neutrophil binding to ICAM-1”.

6) The discrepant findings between the present manuscript by the Vestweber lab and the published Wang et al., 2014 report are not well handled. Firstly, the authors should show their in vitro transmigration data though ICAM-1-Fc coated transwell filters in their manuscript. Secondly, their arguments regarding the values and pitfalls of the different in vivo extravasation models used in the two studies should be better discussed. They have a strong argument that the 'thioglycollate' trigger may produce a non-physiological response (storm) that overrides effects caused by the PILR-α -deficiency. It would have been wonderful if the authors would have shown the 'thioglycollate' and the 'TNF/cremaster muscle' in their paper side by side. Although this would have been the best argument for their findings, they should at least improve their discussion.

We have now discussed these points accordingly in a new in the Discussion section. We have also included the results of the transmigration assay through ICAM-1-Fc coated transwell filters in the Figure 7—figure supplement 3.